# Unifying Specialized Visual Encoders for Video Language Models

## Abstract

The recent advent of Large Language Models (LLMs) has ushered sophisticated reasoning capabilities into the realm of video through Video Large Language Models (VideoLLMs). However, VideoLLMs currently rely on a single vision encoder for all of their visual processing, which limits the amount and type of visual information that can be conveyed to the LLM. Our method, *MERV*, Multi-Encoder Representation of Videos, instead leverages multiple frozen visual encoders to create a unified representation of a video, providing the VideoLLM with a comprehensive set of specialized visual knowledge. Spatio-temporally aligning the features from each encoder allows us to tackle a wider range of open-ended and multiple-choice video understanding questions and outperform prior state-of-the-art works. MERV is up to 3.7% better in accuracy than Video-LLaVA across the standard suite video understanding benchmarks, while also having a better Video-ChatGPT score. We also improve upon SeViLA, the previous best on zero-shot Perception Test accuracy, by 2.2%. MERV introduces minimal extra parameters and trains faster than equivalent single-encoder approaches by parallelizing the visual processing. Finally, we provide qualitative evidence that MERV successfully captures domain knowledge from each of its encoders. Our results offer promising directions in utilizing multiple vision encoders for comprehensive video understanding.

## 1 Introduction

Inspired by the sophisticated reasoning abilities of recent Large Language Models (LLMs) (8; 9; 41), researchers have focused on using them in many other domains to great success. The video counterparts, known as Video Large Language Models (VideoLLMs) (4; 28; 31; 37; 39; 62), connect pretrained vision encoders to LLMs by training a modality bridge from the vision space to the language space, allowing for reasoning to happen in the highly expressive language domain.

Most multimodal LLMs, such as LLaVA (34) for images and Video-LLaVA (31) for videos, opt for contrastively pretrained encoders like CLIP (45) and LanguageBind (70). Their vision-language pretraining naturally lends itself as a bridge between the vision input and the LLM, circumventing the need to train heavy vision-language alignment modules like a QFormer (27). These encoders are almost always pretrained separately and vary in architecture, training data, and optimization strategy. Consequently, the features extracted by these encoders exhibit unique characteristics, each with inherent strengths and limitations. Contrastive encoders like CLIP (45) may be better suited with their multimodal semantic alignment, but are inferior to models such as DINOv2 (42) at fine-grained object level understanding. They also fail to take advantage of models trained specifically on videos, such as ViViT (2). Despite this clear tension between vision backbones, previous research in VideoLLMs has relied on *only* one vision encoder for visual processing as one was thought to be sufficient for visual understanding, and already difficult enough to achieve vision-language alignment with. Any more encoders was unnecessary and not an effective tradeoff of runtime for compute.

In this paper, we argue that this choice to not use multiple encoders in existing VideoLLMs unnecessarily restricts their capabilities. For example, in Figure 1 we can see cases where only one of four different single-encoder models answers a given question correctly. While simple scene descriptions can be answered by image-level models, other questions require temporal and action-level comprehension, benefiting from features encoded with video models like ViViT (2). Consequently, the reasoning capabilities of these VideoLLMs are directly limited by the inherent weaknesses of their respective pretrained encoders. Therefore, employing multiple encoders could allow us to complement one

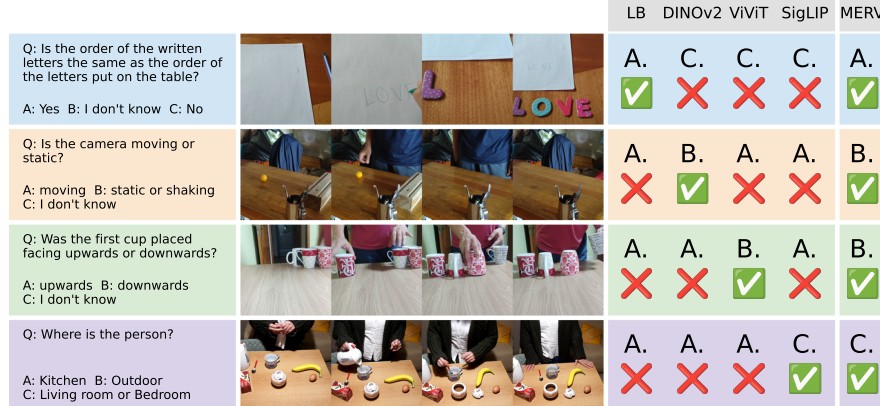

Figure 1: **Different visual experts exhibit individual strengths.** We show some examples where one single encoder model is the only model to correctly answer the Perception Test question (44).

encoder's weaknesses with another encoder's strengths. The wide adoption of the LLaVA paradigm is also indication that vision-language alignment is simple to achieve, even without language-aware vision models.

We propose *MERV*, a Multi-Encoder Representation of Videos, as a new method for integrating multiple visual encoders into a single VideoLLM using a cross-attentive encoder mixer for fusing representations. We introduce a spatio-temporally aligned representation for mixing the information from multiple types of visual encoders. Given the computational complexity of video tasks, we carefully experiment with optimization strategies and parallelizing the visual experts, allowing us to combine four distinct visual encoders with minimal computational overhead. Our frozen method outperforms all of the individual encoder methods, up to 3.7% better than prior works (31) on video reasoning benchmarks, i.e., from 47.1% to 50.8% on ActivityNet-QA (63), and on par with the state-of-the-art (62) on Perception Test (44), a challenging perception and reasoning diagnostic for video models. Finetuning the full model improves MERV past SeViLA (62) by 2.2%, from 46.2% to 48.4%. Finally, we do a detailed qualitative study of our model's capabilities on the Something-Something v2 dataset (15). We show that MERV can accurately capture both the contrastive encoders' (64; 70) strengths on general vision-language understanding, as well as ViViT's (2) specialty on temporally-sensitive tasks (e.g. distinguishing pushing left vs. right), without trading off performance between these specializations as single encoder models do. [1]

## 2 RELATED WORKS

**VideoLLMs** build upon the powerful reasoning capabilities of LLMs by utilizing them as language decoders to enable instruction-followed video understanding. Key advancements include VideoChat (28) and Video-LLaMA (65) for chat-based video understanding, LLaMA-Adapter (66) for pre-alignment, Valley (37) with multilingual LLMs, InternVideo (56) with a dedicated video encoder training phase, and Video-ChatGPT (39) combining video-adapted encoders with LLMs. GPT4Video (57) supports video understanding and generation, while MovieChat (47) focuses on long video comprehension. Models like Chat-UniVi (20) and LLaMA-VID (30) optimize token usage for video representation. Other notable models include Vamos (55), which flexibly uses visual embeddings, action labels, and video captions as input; VideoChat2 (29), developed through three-stage progressive training; Video-LLaVA (31), which aligns image and video representations before projecting them to the LLM space; and VideoPrism (68), which also further trains a video encoder through masked distillation. Specialized models like VTimeLLM (18) focus on fine-grained video moment understanding and time-bound reasoning, while models like Elysium (54) and Merlin (61) can predict object trajectories. SeViLA (62) uses LLM for frame localizer of the video for multiple-choice tasks. Finally, recently LLaVA-Hound-DPO (67) explored using DPO and a higher quality training set for better instruction

---

[1]Our code and pretrained weights will be made public for the camera-ready version of this paper.

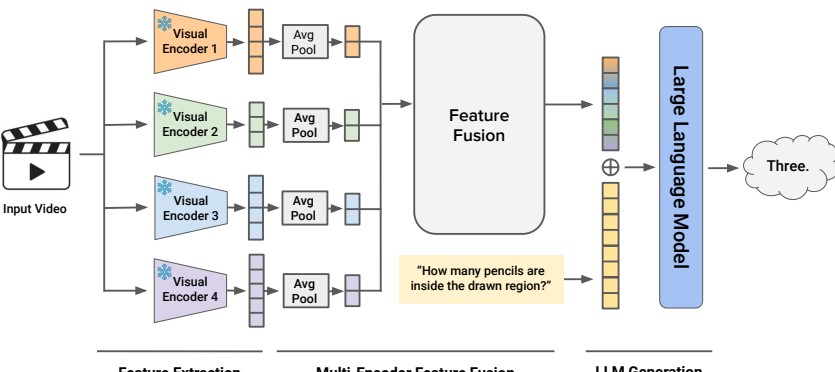

Figure 2: **Overview of MERV, a Multi-Encoder Representation of Videos.** MERV proceeds in three main stages. First, we feed in our input video into each of visual encoders to get different representations. They are then spatio-temporally aligned before being fused by a cross-attentive mixer. The output is a visual embedding with an additive mix of information from all of the encoders, which is combined with the text query to produce our final generation.

following. Distinct from these aforementioned works, our approach centers on utilizing a diverse array of visual encoders, each with its own unique strengths and especially a video encoder, to significantly enhance the capabilities of the VideoLLM framework. By strategically utilizing these specialized encoders, we aim to capture a broader spectrum of visual information, thus enriching VideoLLMs' understanding of video content.

**Combining multiple encoders for multimodal LLMs** is gaining attention. Eyes Wide Shut (51) explored mixing DINOv2 and CLIP features for LLaVA, but their results signal that mixing features effectively requires investigation. Both Mipha (71) and Prismatic-VLMs (22) found that image encoders like CLIP and SigLIP, which are trained using vision-language contrastive loss, surpass other image encoders such as ViT and DINOv2, with SigLIP showing further improvements over CLIP. SPHINX-X (14) and SPHINX (32) combines multiple image encoders by concatenating features along the channel dimension, while BRAVE (21) concatenates features from multiple encoders sequence-wise, followed by a QFormer with masked modeling. There is also the popular body of research on multimodal LLMs using many modalities including image, video, audio and/or 3D (7; 16; 17; 25; 33; 36; 38; 43; 48; 49; 65). In contrast, this paper dives into the video-language domain, exploring combining multiple image and video encoders and exploiting their structural similarities. Our feature fusion is both performant and efficient in FLOPs, and results in an all-encompassing additive mixture of features which previous works were unable to create without tradeoffs.

## 3 MERV: MULTI-ENCODER REPRESENTATION OF VIDEOS

Our goal for MERV is to systematically build a video model that leverages multiple encoders with an LLM to process a video following the LLaVA/PrefixLM (34; 35) paradigm (see Figure 2). Unlike previous works, our focus is not on combining multiple *modalities* (3; 70), but instead on combining multiple image and video encoders trained on different datasets and objectives. We extensively ablate three key aspects to make this possible: our selection of *multiple* encoders, i.e., which visual encoders and how many to use (Sec 3.1); how we *align* the spatio-temporal representations of each encoder to mix the information together, especially in an efficient manner (Sec 3.2); and our *implementation* efficiencies, from the parallel visual processing to the training recipes (Sec 3.3).

### 3.1 MULTI-ENCODER FEATURE EXTRACTION

Our final architecture uses four distinct types of models: spatial experts, fine-grained temporal experts, image-language experts, and video-language experts. We found experimentally that our choice of four performed the best across all types of questions, and ablate our choices in Section 4.2. More details about these four encoders and other encoders we considered are in Appendix Table 4.

**Spatial expert:** DINOv2 (42) is trained using unsupervised learning on local-to-global correspondences in image data. The resulting features have a robust understanding of object parts, as well as semantic and low-level image understanding, but can suffer from poor language grounding.

**Temporal expert:** ViViT (2) is trained using supervised learning on short videos. The architecture is designed for modeling the interactions between frames using spatial and temporal attention, which lets it capture longer temporal dependencies than pure image models can.

**Image-Language contrastive expert:** SigLIP (64) is trained using sigmoid contrastive learning on image-text pairs. The model is designed to learn a joint embedding space for images and text, which makes it good at understanding vision-language associations. However, it can overlook the finer details of an image which are not well described by text in its training data.

**Video-Language contrastive expert:** Finally, our video-language expert is LanguageBind (70). Used by Video-LLaVA (31), LanguageBind is trained through joint multimodal learning between text and multiple modalities, including videos, infrared, and audio, and understands the relationship between video and text and their high-level semantics. We only use the video encoder of LanguageBind.

### 3.2 SPATIO-TEMPORALLY ALIGNED REPRESENTATIONS FOR FEATURE FUSION

Our input is a batch of text, image-text, or video-text queries. The visual part of the input, either images $\mathcal{I}$ or videos $\mathcal{V}$, is passed through each of the visual encoders to extract the respective features. Here we describe the detailed care we took in pre-processing to prepare the features for alignment.

First, images are treated as videos with repeated frames, so assume all inputs are videos from here on out. A video is of shape $T \times H \times W$, where $T$ is the number of frames and $H, W$ are the height and width of the frames, and produce an output of shape $t_e \times h_e \times w_e$ for an encoder $e$. One obstacle with using different visual encoders is that each model outputs features with a different structure. For example, given an input of shape $16 \times 224 \times 224$, ViViT outputs a feature of shape $8 \times 14 \times 14$ whereas LanguageBind's features are of shape $16 \times 16 \times 16$. Image-based encoders will not change the temporal dimension, whereas ViViT downsamples the frames by a factor of 2.

For temporal alignment, as each encoder is flexible enough to handle varying input frames, we simply choose our input $T$ for each encoder so that each output $t_e$ is the same across all encoders, i.e. $t$.

**Pre-fusion projection.** Now we need to achieve spatial alignment among the features. Naïvely combining them is not possible as they all have different spatial shapes, and would also be prohibitively expensive at full resolution. We design a pre-fusion projector to both align and compress them.

Suppose our feature from encoder $e$ is $\mathbf{v}_e \in \mathbb{R}^{t \times h_e \times w_e \times d_e}$, where $d_e$ is the dimension of encoder $e$, and assume the output spatial representations are square (i.e. $h_e = w_e$, but we keep notation for clarity). Our pre-fusion projector uses an adaptive 2D average pool $\mathcal{P}$ for each encoder to resize the spatial dimensions to the same $h \times w$ for all encoders, where $h < h_e$ and $w < w_e$. As $t$ is the same across each $\mathbf{v}_e$, this *spatio-temporally aligns* the representations.

Finally, we need to connect the varying embedding dimensions $d_e$ to a same dimensional space. We add a linear layer to project the features from dimension $d_e$ to $d$, the LLM's dimension. In total, our pre-fusion projection is

$$\mathbf{x}_e := \mathcal{P}(\mathbf{v}_e)W_e \in \mathbb{R}^{\ell \times d} \quad \text{for } e \in \text{Encoders} \tag{1}$$

where $W_e \in \mathbb{R}^{d_e \times d}$ is each encoder's output linear layer, and $\ell = t \times h \times w$. This projector is lightweight, having only $d \times \sum_e d_e$ trainable parameters for dimension matching, making it easy to scale to an arbitrary number of visual encoders. For detailed ablations, see Section 4.2.1.

**Feature fusion strategies.** The final part of our pipeline is fusing the multi-encoder information together using cross-attention with learnable queries to additively mix the different representations together. The visual features determine the weights of the linear mixture, which we find sufficient for our task. We use a single randomly initialized query $\mathbf{Q} \in \mathbb{R}^{1 \times d}$, keys as $\overline{\mathbf{X}} = [\overline{\mathbf{x}}_1 \ \ldots \ \overline{\mathbf{x}}_N] \in \mathbb{R}^{N \times d_L}$, where $\overline{\mathbf{x}}_e \in \mathbb{R}^d$ is each encoder's features averaged over the sequence dimension $\ell$ for a faster computation, and $N$ the number of encoders, and values as $\mathbf{X} = [\mathbf{x}_1 \ \ldots \ \mathbf{x}_N] \in \mathbb{R}^{N \times \ell \times d}$. We calculate our final unified feature as

$$\mathbf{O} := \mathrm{Softmax}\left(\frac{\mathbf{Q}\overline{\mathbf{X}}^{\top}}{\sqrt{d}}\right)\mathbf{X} \in \mathbb{R}^{\ell \times d}. \tag{2}$$

The final step is to concatenate the visual embedding and tokenized text together into the LLM. We use the base LLaMA-2 7B model (53), which we found performs better than the chat model. We test multiple alternate feature fusion strategies and their tradeoffs in Section 4.2.2.

### 3.3 IMPLEMENTATION EFFICIENCIES

**Parallelized visual encoding.** At a first glance, using multiple encoders seems to be a large cost to pay when comparing the raw FLOPs and parameters. However, a key benefit of the LLaVA style architecture is that the entire feature extraction and projection pipeline can happen *in parallel*. To make this possible, we build on top of the recent powerful advances in parallel processing for LLMs and use PyTorch's Fully Sharded Data Parallel (69). As the video encoders themselves are much smaller than the LLM blocks and complete in around the same time, most of the overhead in running four encoders is already covered by having one encoder. We provide some timing numbers in Section 4.2.3 and find that our step time is similar to that of the single-encoder methods.

Our code is built on top of the Prismatic VLM codebase (22), which efficiently implements vision-language model (VLM) training. We add the ability to handle videos and an arbitrary number of visual encoders, along with many useful features for training. Our training is efficient for using multiple visual models, completing in under 24 hours using 8 L40-48GB GPUs, and down to 8 hours using 8 H100s in limited access testing. The Video-LLaVA codebase runs Stage 2 in around 38 hours on the same L40 setup and could not easily support multiple encoders in our initial attempts.

**MERV frozen and full.** Many different recommendations for training LLaVA style models have been made since its inception. This is only made more complicated by the introduction of new datasets with every new VideoLLM architecture, making it difficult to properly determine the best recipe for one's own setup. We intentionally fix our dataset to be the same as Video-LLaVA's so we can isolate the impacts of the training setup, from which we find two viable settings. *MERV (frozen)*, which performs only Stage 2 instruction tuning and achieves similar results to the original Video-LLaVA recipe in only 43% of the time, and *MERV (full)*, which unfreezes the LLM during Stage 1 as well for a slight improvement on a few benchmarks. As MERV (frozen) is faster to train with similar performance, we adopt that recipe by default for analysis, and interchangeably use *MERV* to refer to it for simplicity from here on out. Detailed analysis is provided in Section 4.2.3.

## 4 EXPERIMENTAL RESULTS

**Datasets and training procedure.** Our data mix is the same as Video-LLaVA (31). The Stage 1 data is single-turn concise captioning, with 558k (image, text) pairs from LAION filtered by LLaVA (34) and 702k (video, text) pairs from Valley (37). The Stage 2 data is multi-turn conversations, detailed captioning and reasoning, with 665k (image, text) pairs from LLaVA (34) and 100k (video, text) instructions from Video-ChatGPT (39).

All the preprocessing, including frame extraction, adheres to the original method that each encoder is trained with. We extract 16 uniformly sampled frames from each video, except for ViViT which extracts 32 frames by default but produces a 16-frame output feature.

For MERV (frozen), we train on only Stage 2 data for 1 epoch with a learning rate of $2 \times 10^{-5}$ and a batch size of 128 with gradient accumulation. For MERV (full), we first train on Stage 1 data with a learning rate of $1 \times 10^{-4}$ and the projectors, feature fusion, and LLM unfrozen with similar settings. Both recipes use an initial warmup ratio of 0.03 and a cosine schedule.

**Evaluation.** We evaluate our model on a comprehensive suite of video understanding benchmarks, including the open-ended MSVD-QA (59), MSRVTT-QA (59), TGIF (19), and ActivityNet-QA (63), as well as the multiple-choice benchmarks NExT-QA (58), VLEP (24), TVQA (23), and Perception Test (44). We emphasize that NExT-QA, VLEP, and TVQA datasets are **held-out** datasets that we did not use during our experiments, and only evaluated once after all the design is completed. We report both accuracy and score following the Video-ChatGPT evaluation protocol (39) where

| Methods | Visual Encoder And LLM | MSVD-QA | | MSRVTT-QA | | TGIF-QA | | Perception | ActivityNet-QA | | NExT-QA | VLEP | TVQA |
|---|---|---|---|---|---|---|---|---|---|---|---|---|---|
| | | Acc | Score | Acc | Score | Acc | Score | Acc | Acc | Score | Acc | Acc | Acc |
| *Alternative data mixes* | | | | | | | | | | | | | |
| Video-Chat (28) | (50), (11) | 56.3 | 2.8 | 45.0 | 2.5 | - | - | - | 26.5 | 2.2 | - | - | - |
| LLaMA-Adapter (66) | (45), (52) | 54.9 | 3.1 | 43.8 | 2.7 | - | - | - | 34.2 | 2.7 | - | - | - |
| Video-LLaMA (65) | (50; 26), (8) | 51.6 | 2.5 | 29.6 | 1.8 | - | - | - | 12.4 | 1.1 | - | - | - |
| Video-ChatGPT (39) | (45), (8) | 64.9 | 3.3 | 49.3 | 2.8 | - | - | - | 35.2 | 2.7 | - | - | - |
| SeViLA (62) | (50), (26) | - | - | - | - | - | - | 46.2 | - | - | **63.6** | **64.4** | 38.2 |
| LLaMA-VID-7B* (30) | (13), (8) | 69.30 | 3.74 | 57.84 | 3.24 | 51.31 | 3.26 | 41.64 | 46.45 | 3.22 | 60.61 | 57.65 | 37.43 |
| LLaMA-VID-13B* (30) | (13), (8) | 70.25 | 3.77 | 58.58 | 3.26 | 51.26 | 3.26 | 41.54 | 46.79 | 3.23 | 60.03 | 61.98 | 41.33 |
| *Same data mixes* | | | | | | | | | | | | | |
| Video-LLaVA* (31) | (70), (8) | 67.74 | 3.69 | 56.90 | 3.18 | 47.99 | 3.17 | 44.22 | 47.08 | 3.27 | 59.61 | 61.21 | 37.66 |
| MERV (frozen) | (70; 42; 2; 64), (53) | **70.97** | 3.76 | **59.03** | **3.25** | 51.1 | 3.26 | 46.21 | **50.87** | **3.34** | 63.09 | 58.66 | **42.28** |
| Gains to Video-LLaVA* | | +3.23 | +.07 | +2.13 | +.07 | +3.11 | +.09 | +1.99 | +3.79 | +.07 | +3.48 | -2.55 | +4.62 |
| MERV (full) | (70; 42; 2; 64), (53) | 70.48 | **3.79** | 57.25 | 3.24 | **51.39** | **3.28** | **48.41** | 49.93 | 3.33 | 61.36 | 60.07 | 39.42 |
| Gains to Video-LLaVA* | | +2.74 | +.10 | +0.35 | +.06 | +3.40 | +.11 | +4.19 | +2.85 | +.06 | +1.75 | -1.14 | +1.76 |

Table 1: **Comparison of different multimodal LLMs on video reasoning benchmarks**. We employ ChatGPT to evaluate performance following Video-ChatGPT (39) where applicable (version `gpt-3.5-turbo-0613`). * denotes our evaluation of using the author provided checkpoint. The first five datasets were used as development sets; the last three were held-out for our final evaluation.

applicable, and all evaluations are done zero-shot without any dataset-specific fine-tuning. Results using GPT-3.5-turbo for evaluation are done with the June 13th, 2023 cutoff date.

## 4.1 Comparison to State of the Art

Table 1 tabulates the performance of MERV (frozen) and (full). We compare our model to the existing state-of-the-art works, including Video-LLaVA (31) that share our training data mixture, and other VideoLLMs (28; 30; 39; 62; 65; 66). We find that our method, generating video representations using multiple visual encoders that specialize in different skills of video understanding, outperforms Video-LLaVA across nearly all of the benchmarks, with a 3.2% gain on MSVD and a 3.7% gain on ActivityNet. Both of our methods perform better overall than Video-LLaVA, even when using less data with just Stage 2 as shown by the MERV numbers. While MERV (full) is not a strict improvement to MERV, it still improves on some difficult benchmarks with its additional video-language alignment. We believe that outside of these testing sets, MERV (full) is a better model overall and recommend using this recipe when possible. Compared to LLaMA-VID (30), which uses a different training mix, we also better in nearly all benchmarks, up to around 4.5% across Perception Test, ActivityNet, and TVQA. [2] MERV (full) outperforms the previous state-of-the-art on the Perception Test zero-shot with 48.4%, compared to SeViLa (62) with a 46.2% accuracy. Overall, our design shows a significant improvement over Video-LLaVA and prior methods as a whole.

## 4.2 Ablations

In this section, we justify the design choices for our our architecture, covering our projectors, feature fusion strategies, and training recipes. Our ablations are done with the MERV (frozen) recipe.

### 4.2.1 Pre-Fusion Projectors

The first module we investigate is our projectors, which serve to connect each encoder from its pretrained embedding space to a common embedding space.

We test two types of projectors: image-level, which operate on frames independently, and video-level, which aggregate information across frames. The image-level projectors are similar to those described in MM-1 (40): 2D adaptive average pooling, a shallow attention resampler similar to a Perceiver Resampler (1), and convolutional pooling with 3 RegNet blocks on both sides of an average pool layer like the C-Abstractor in Honeybee (6). For video-level projectors, we use a 3D average pool, where we pool to the same spatial dimension but furthermore pool the frame dimension by 2, and a 3D convolution where we add a single $2 \times 3 \times 3$ convolution before the same average pooling. For all projectors, we project to the same number of tokens $t \times h \times w$, using an adaptive average

---

[2]Video-ChatGPT's and Video-LLaVA's author-reported numbers on TGIF are incomparable as they were on a subset of the dataset. See https://github.com/PKU-YuanGroup/Video-LLaVA/issues/37.

| Projector | Avg Acc | Params | FLOPs |
|---|---|---|---|
| 257 tok | 54.76 | - | - |
| class tok | 52.05 | - | - |
| 2D Avg | 54.96 | 0 | 2.1M |
| 2D Avg* | **55.86** | 0 | 4.2M |
| 2D Attn | 52.12 | 12.7M | 9.7G |
| 2D Conv | 54.23 | 237M | 241G |
| 3D Avg* | 55.09 | 0 | 4.2M |
| 3D Conv | 55.42 | 113M | 232G |

(a) **Pre-fusion projectors.** * is 16 frames instead of 8. Top two rows are projector-free baselines.

| Tkns | MSVD | MSRVTT | TGIF |
|---|---|---|---|
| 1 | 61.94 | 54.64 | 41.41 |
| 4 | 64.47 | 55.72 | 45.32 |
| 16 | 67.23 | 56.44 | 47.75 |
| 64 | **69.08** | **58.00** | **50.01** |
| 100 | 68.38 | 57.47 | 48.78 |
| 144 | 68.65 | 57.73 | 48.81 |
| 256 | 68.46 | 57.72 | 48.66 |

(b) **Pre-fusion output token.** We ablate the optimal token size per frame for the pre-fusion projector.

| Strategy | Avg Acc | FLOPs |
|---|---|---|
| Cross-Attn | **56.83** | 17.19 T |
| Concat (Seq.) | 54.45 | 43.09 T |
| Concat (Ch.) | 56.64 | 16.29 T |
| Learnable W | 55.01 | 16.24 T |
| 25% - Mixed | 54.19 | 16.39 T |

(c) **Feature fusion strategy.** Cross-Attn additive mixing is best overall among all the strategies on accuracy, for its FLOPs.

Table 2: **Ablating design choices.** We highlight our defaults in orange and **bold** the best results. Average accuracy is on MSVD, MSRVTT, TGIF, and Perception Test. Full results are in the Appendix.

pool or $h \times w$ latent tokens for the attention resampler. We report average performance across our development sets of MSVD, MSRVTT, TGIF, and Perception Test.

**Pre-fusion projector.** Table 2a tabulates each projector's average accuracy, along with their parameter count and FLOPs, with LanguageBind as the single vision encoder and an 8 frame 64 token projection output by default. We find that 2D average pooling is the best overall projector, surpassing that of the full 257 token embedding (used in Video-LLaVA (31)) while also having no trainable parameters and the fewest FLOPs. The projection serves as a form of feature selection, allowing the LLM to efficiently reason only over the most relevant information. However, increasing the frame resolution from 8 to 16 was a large improvement, showing that increasing temporal resolution is still important. One result worth noting is the poor performance of the attentive resamplers, typically a popular projector choice. They are agnostic to structure, which leads them to being weaker projectors for us. This highlights the importance of aligning representations with their spatial and temporal structure, *especially* for video models, which extract many more frames of visual information.

**Projector token length.** Similarly, we ablate the optimal output token size of the projector. Table 2b tabulates performance on different token output sizes when using 2D Average Pooling with 16 frames as the projector. We see that the performance peaks at 64 tokens, with worse performance for longer token lengths. This balances the number of tokens used for condensing the visual embedding while also minimizing the extra processing needed by the LLM, leading to the best overall performance.

### 4.2.2 FEATURE FUSION STRATEGIES

Next, we test different strategies for fusing the information from all of the features, with detailed breakdowns in Table 2c. First, we evaluate two popular concatenation methods, in either the token sequence dimension, or the channel dimension followed by an MLP projector for matching the LLM dimension. While sequence-wise concatenation is widely used in multimodal LLMs (51), our method outperforms it while using significantly less computation, with a 56.8% average accuracy compared to 54.4%, while also using 2.5× fewer FLOPs. Concatenation channel-wise reaches a similar performance of 56.6% and a lightweight cost. However, our cross-attention shows slightly better performance, with the additional benefit of having accessible encoder weightings for analysis, so we do not choose channel-wise concatenation as our final design. We also try different methods of additive mixing as an ablation. The last two rows of Table 2c show the performance when either learning the additive weights directly as a learnable scalar or by fixing the weights to be 0.25 for each of 4 encoders. We see that using cross attention outperforms both methods by 1.8% and 2.6%, as our feature fusion module can dynamically generate better fused embeddings given the visual input.

### 4.2.3 TRAINING RECIPES

Finally, we also compare different training recipes based on the literature and our own expertise. Traditional rule-of-thumb follows that of the original LLaVA recipe: a Stage 1 pre-training on captioning data to align the projectors only, and a Stage 2 instruction tuning on multi-turn complex reasoning data for both projectors and the LLM. Many recent works have attempted some combination of other strategies, such as unfreezing the vision encoders (12) or skipping the Stage 1 (22).

We systematically map out this landscape, fixing our dataset to be the same as Video-LLaVA's and testing multiple hypotheses for the video domain. Contrary to Video-LLaVA, we found that the Stage 1 phase did not have a significant effect on the final performance when training only the projectors and feature fusion, as can be seen in Table 5 in the Appendix. Performing only Stage 2 instruction tuning leads to similar results in 43% of the total time, so we adopt this recipe for efficiency. The general performances fluctuate, with the Perception accuracy of the Video-LLaVA recipe being slightly higher by 1.2%, but MSVD, MSRVTT, and ActivityNet of ours are higher by around 0.4%. We refer to this recipe as *MERV (frozen)*.

This recipe is still unsatisfying as it leaves a large amount of data, approximately 1.3M vision-text pairs, unused for training. In our empirical observations, we often found that video-language alignment was not very strong. The distributions of language used in video datasets and benchmarks seem to sparsely overlap based on their sentence embeddings, which could have impacted our ability to perform well zero-shot on the downstream benchmarks. However, we found that if we *unfreeze* the LLM during Stage 1 and learn alignment between the LLM and the projectors and feature fusion, our performance improved on a few key benchmarks, especially Perception Test, by up to 2.2%.

As another ablation, we train MERV on a single stage comprised of the Stage 1 and Stage 2 data mixed together (bottom of Appendix Table 5). Surprisingly, this does worse than the explicit two stage training recipe. We attribute this to the explicit types of data in each stage being a form curriculum learning, showing that these stages are still important for optimal performance.

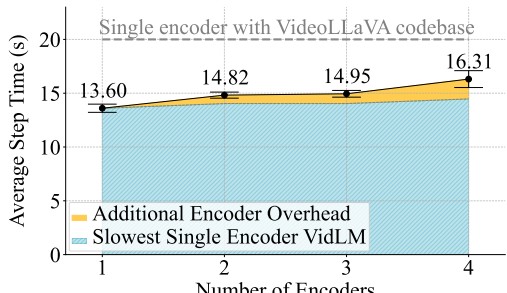

Finally, we provide evidence for the efficiency of our method. We use the default FSDP sharding strategy PyTorch provides; it is not currently possible to specify explicit plans for which modules go where (but may be possible as FSDP matures). However, even with this basic strategy, our method is dominated by the slowest single encoder present, incurring very little additional overhead from extra encoders due to this parallelization, making it cost efficient to scale up in the number of encoders.

Figure 3: **Extra encoders incur minimal step time overhead**. Here we add encoders in the order of DINOv2, LanguageBind, SigLIP, ViViT, plotted alongside the slowest single encoder in each group.

## 5 ANALYSIS

### 5.1 STRENGTH IN THE ENSEMBLE OF ENCODERS

The original motivation of our work was to choose encoders with complementary visual knowledge to form a comprehensive representation for our final model. The key questions are 1) do we benefit by using more than one encoder, and 2) do we need all four encoders, i.e. does each one meaningfully contribute to the final performance?

**Can we make use of more encoders?** The conventional wisdom is to use a single encoder, typically a contrastively trained vision-language model like CLIP, SigLIP, or LanguageBind (45; 64; 70), in a VideoLLM. In Figure 4a, we show the four single encoder models corresponding to each of our chosen encoders using their full embeddings. They not only all perform worse than MERV but also use more FLOPs, as without our pre-fusion projectors, their sequence lengths are at least 4× ours.

**Are each of the encoders contributing?** To affirm that this set of four encoders is actually beneficial for improving understanding, we train three-encoder VideoLLMs under the same strategy, but removing a different encoder each time. Each of these models does worse based on the strength of the encoder removed, meaning that MERV is using their knowledge (Fig. 4a). The minor drop in FLOPs illustrates how most of the computation is still dominated by the LLM, not the vision encoders.

**Does MERV capture visual skills of different encoders?** Finally, we ask if our model effectively captures knowledge from its encoders. We first answer through our previous open-ended QA benchmarks. To assess the performance across different visual tasks, we create "pseudo"-skill

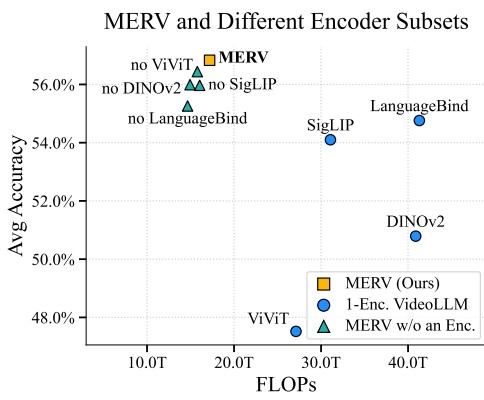 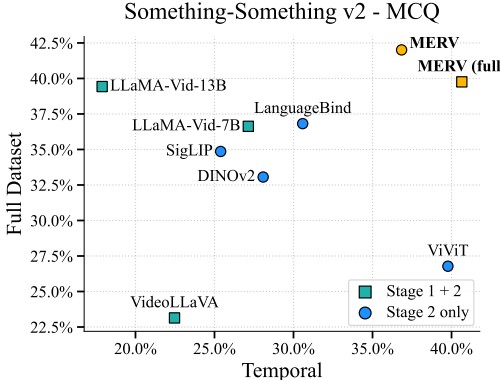

(a) **Visual Encoder Subsets.** MERV outperforms single-encoder VideoLLMs, with our feature projectors unlocking more computational effiency. Removing any encoder also reduces MERV performance.

(b) **SSv2-MCQ and Temporal.** Temporal denotes performance on 12 selected classes where actions are indistinguishable if played in reverse.

Figure 4: **Analysis plots supporting our design of multiple encoders, from their accuracy to their skill specializations.** Average accuracy is across MSVD, MSRVTT, TGIF, and Perception Test. Full results are in the Appendix Tables 7, 8, 10.

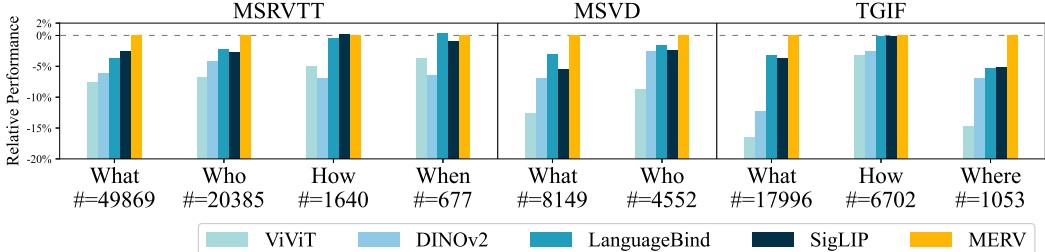

Figure 5: **Single encoder vs. MERV on different types of video tasks**. We plot the relative performance of VideoLLMs with different visual encoders. While each single encoder has its strength in different tasks, our method shows better performance than all the other single encoders in almost every task. We only plot tasks with more than 500 samples. See Appendix for details.

categories by looking at the first word of the question sentence, which are often WH-words. They can be viewed as a proxy of skills required to solve the task. For example, `Where` requires spatial understanding and `When` requires temporal understanding. Figure 5 shows the relative performance of different visual encoders. While the contrastive models generally dominate each category, no single encoder performs best in all tasks. LanguageBind, for example, performs the best in TGIF-`What` with 46.23%, while DINOv2 performs on par with the best in MSVD-`Who` with 82.12%. Our method which combines different encoders into an unified representation that consistently matches or improves the best-performing encoder. Raw numbers are in Table 9 in the Appendix.

## 5.2 MERV CAN INTUIT MOTION AND GENERAL UNDERSTANDING SIMULTANEOUSLY

We take an alternate angle to quantifying how well our model learns from each of its individual encoders by looking towards classic video action recognition, from which we create detailed categories of skills based on the class names. We turn to Something-Something v2 (15) (SSv2) dataset where the goal of the original benchmark is to classify the video into one of 174 classes, e.g., *Pulling [something] from left to right*. This allows us to analyze our model's understanding of temporal-spatial interaction with minimal distractions from scene understanding and real-world semantics. We provide qualitative examples where MERV is able to simultaneously provide descriptions and motion understanding, where it tends to align with either SigLIP (first row) or SigLIP (second row) based on the task (Figure 11). However, evaluating SSv2 as a zero-shot VideoLLM task is difficult with many specific categories. We repurpose the dataset as a 5-choice multiple-choice question (MCQ) dataset

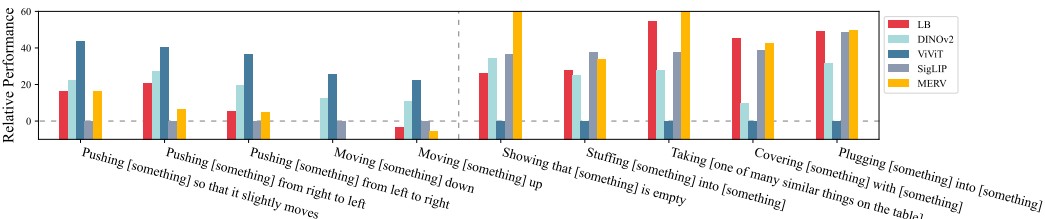

Figure 6: **Single-Encoder Performance Difference in Something-Something v2 - MCQ**. ViViT shows better performance on tasks where temporal understanding is crucial, while LanguageBind and SigLIP show better performance where task can be solved from single-frame understanding.

and fix the prompt to be "How is the object in the video being interacted with?". Incorrect choices were randomly sampled from the 173 other classes. We call this benchmark "Something-Something v2 - MCQ" to distinguish it from the original classification task.

We also selected 12 classes *a priori* from SSv2, where the action is indistinguishable if reversed in time, e.g., *Pulling [something] from left to right* and *Pulling [something] from right to left*. Figure 4b plots the performance of MERV and single-encoder models on this temporal subset ($x$-axis) against the full dataset ($y$-axis). We see that ViViT, which often falls short in other Video QA benchmarks, surprisingly performs better than other encoders at 39.77%, which is 9.19% higher than the next closest model LanguageBind. However for the full dataset, ViViT suffers with a worse performance of 26.78%, as ViViT's strength is on temporal understanding despite lacking in vision-language understanding. Contrastive encoders have the upper-hand on most other classes.

We plot the performances of 10 SSv2 classes where the performance difference between ViViT and SigLIP is largest in Figure 6. We see that actions that cannot be inferred from a single frame are the ones that ViViT performs better, e.g., *Moving [something] down* is indistinguishable from *Moving [something] up* if temporal information is omitted. Meanwhile, SigLIP performs better for classes where understanding the semantics of the scene can hint the action that is happening, e.g., if the video contains a cup and a bottle of water, one can easily expect *Showing [something] is empty* without watching the full video. See Appendix Figure 9 for sample videos of the 10 classes.

We believe that the architecture, datasets, and objective of each model causes these difference. ViViT processes spatial-temporal tubelets for embeddings, leading to better temporal understanding despite only being pre-trained on Kinetics-400 classification. SigLIP uses image-based ViT with no temporal layer has limited temporal understanding, but has a greater knowledge due to its larger training set and contrastive objective. MERV, at 42%, shows better performance compared to all these single-encoder models via leveraging strength of all the individual encoders. MERV (full) performs better than both VideoLLaVA (31) and the 7B and 13B variants of LLaMA-Vid (30).

## 6   CONCLUSION

Previous VideoLLMs have been limited to relying on a single visual model for feature extraction, which leads to limited understanding capabilities of vastly different video tasks. In our work, we break this paradigm and explore various fusion strategies for combining information from multiple visual experts to generate a representation that can leverage the capabilities of different video encoders. We find that our multi-encoder feature fusion is able to outperform comparable methods by up to 3.79% on video reasoning benchmarks. We show that the method can obtain better performance than the best-performing single-encoder model with minimal computational overhead. Finally, we quantitatively and qualitatively observe the skill specializations our model learns on an MCQ format of Something-Something v2, which confirms both that encoders can be specialized and that our model captures both axes of knowledge. Our paper proposes some initial steps in rethinking how we approach the use of multiple encoders. We are especially excited about this trend as it could allow our model to scale visual processing with the number of GPUs with better sharding strategies. We can place one expert on each device and obtain visual features in parallel while still retaining similar runtimes to having just one expert. We hope that this inspires others to also consider this problem as potentially another direction for scaling and improving their VideoLLMs.

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

# A  APPENDIX / SUPPLEMENTAL MATERIAL

## A.1  LIMITATIONS AND BROADER IMPACT.

Our works are based on LLaMA-2 7B model (53), and as with many other VideoLLM models, the performance of our method is hugely dependent on the capabilities of the LLM model, and better-performing models often demand significantly more computation. MERV requires running multiple encoders, which can be computationally intensive and can lead to out-of-memory errors in limited resource settings. While FSDP (69) allows us to easily and effectively train larger models across multiple GPUs than would otherwise be possible, its generality also makes it difficult for us to design tailored sharding strategies that would maximize the performance of our model. However, with future improvements to data parallelism, our model can still benefit greatly and run even more efficiently. Also, despite the improved speeds, there is still an upper bound for what constitutes a *reasonable* training time that still allows us to test many of our design assumptions, which limits the scale and number of experiments we can run.

While we show that our method can successfully leverage information from different visual encoders, nevertheless if the encoders themselves are limited in video understanding capability, MERV cannot fully compensate for that. Our work aims to facilitate video understanding, which can lead to positive social impacts such as a video captioning model for low-vision users, automatic detection of medical emergencies, or better self-driving cars. It can also lead to negative social impacts like easy surveillance by the authorities, and human-like internet bots being used for scamming purposes. We follow the same safeguards implemented by the original authors of the datasets, the visual models, and the LLM models. We have not put any additional safeguards ourselves.

## A.2  QUALITATIVE RESULTS

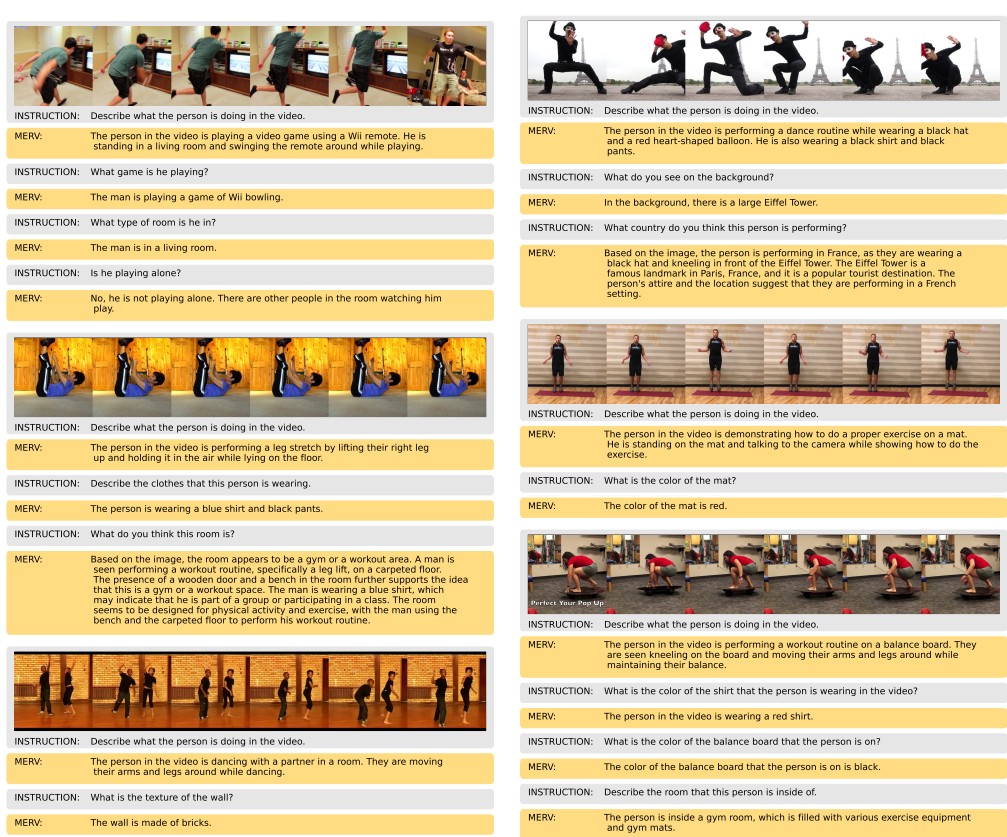

Figure 7: Samples of MERV in video understanding

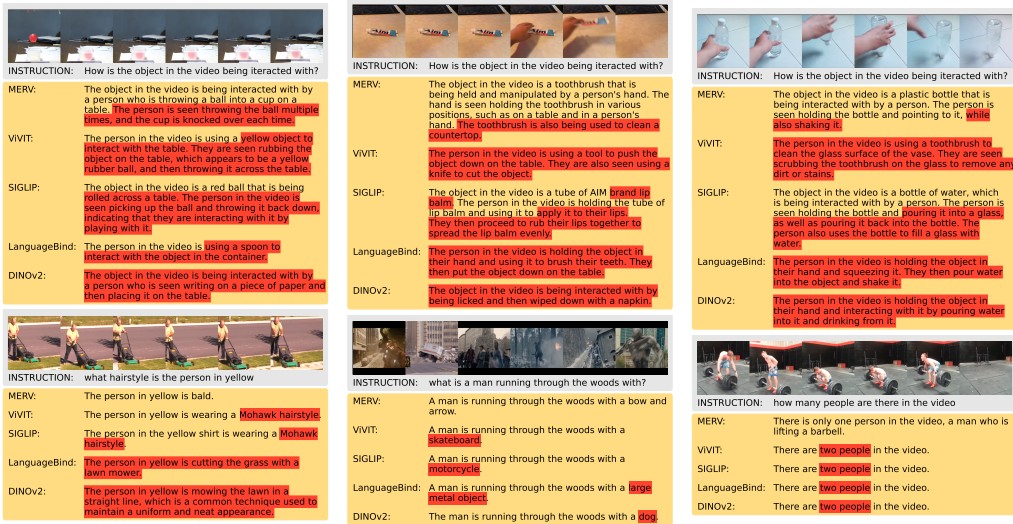

Figure 8: **MERV examples**. MERV tend to show improved understanding in temporal-heavy videos as in Something-Something v2 dataset (15) (Top Row), while retaining the performance on scenic understanding, seen from popular video benchmarks (59; 63) (Bottom Row).

## B  TRAINING DETAIL

### B.0.1  BASELINE ENCODER AND LLM DETAILS

| Model | Visual Encoder | LLM |
|---|---|---|
| Video-Chat (28) | ViT-G (EVA-CLIP) (50) | StableVicuna (11) |
| LLaMA-Adapter (66) | CLIP (45) | LLaMA-1 7B (52) |
| Video-LLaMA (65) | ViT-G (EVA-CLIP) (50) + BLIP-2 Q-Former (26) | Vicuna-7B v0 (8) |
| Video-ChatGPT (39) | CLIP (45) | Vicuna-7B v1.1 (8) |
| SeViLA (62) | ViT-G (EVA-CLIP) (50) + BLIP-2 Q-Former (26) | FlanT5-XL (3B) (10) |
| LLaMA-VID-7B (30) | EVA-G (13) | Vicuna-7B v1.5 (8) |
| LLaMA-VID-13B (30) | EVA-G (13) | Vicuna-13B v1.5 (8) |
| Video-LLaVA* (31) | LanguageBind (70) | Vicuna-7B v1.5 (8) |

Table 3: **Visual Encoder and LLM Information**.

### B.0.2  MERV ENCODER DETAILS

| Model | Architecture | Expertise | Training Datasets | Training Objective |
|---|---|---|---|---|
| LanguageBind (70) | ViT-L/14 | Video+Language | VIDAL-10M, five-modal video examples | Contrastive |
| DINOv2 (42) | ViT-L/14 | Spatial | LVD-142M | Self-Supervised |
| ViViT (2) | ViViT-B/16×2 | Actions/Temporal | Kinetics-400/600, short videos | Supervised |
| SigLIP (64) | ViT-B/16 | Image+Language | 4B curated image/text pairs | Contrastive |

Table 4: **Encoder Information**. Detailed information about the four encoders used in our experiments. They represent a broad coverage of visual information and training objectives.

Here, we detail the visual encoder details, LLM, and the training objectives. We plan to release the code for the camera-ready version of the paper.

**LanguageBind** We use the code from the original author, using the pre-trained weight `LanguageBind/LanguageBind_Video_merge` uploaded on `huggingface`.

**DINOv2** As DINOv2 is an image-model, we get embedding per frame, and concatenate them to be a video embedding. We use `ViT_Large` model, pre-trained on LVD-142M dataset, and take the penultimate layer for the embeddings. Specifically, we use `timm`'s `vit_large_patch14_reg4_dinov2ịvd142m`

**ViViT** We use $\text{ViT}_{base}$ as our backbone, pre-trained on Kinetics-400 dataset. Specifically, we use `google/vivit-b-16x2-kinetics400` uploaded on `huggingface`. We use featurizer output as the video embedding.

**SigLIP** As SigLIP is an image-model, we get embedding per frame, and concatenate them to be a video embedding. We use $\text{ViT}_{base}$ as our backbone, and take the penultimate layer for the embeddings. Specifically, we use `timm`'s `vit_base_patch16_siglip_224`

We also considered multiple other options for encoders, such as CLIP-ViP (60) for our video-language contrastive expert, V-JEPA (5) and Hiera (46) for our pure video model, and CLIP (45) for our image-language contrastive expert, but found that our choices performed better overall.

## B.1 DETAILED EXPERIMENTAL RESULTS.

Here we tabulate the full experimental results that was abbreviated from the main paper. The first table (Table 5) ablates the different training recipes we tried for MERV, with extended discussion in Section 4.2.3.

| Methods | MSVD-QA | | MSRVTT-QA | | TGIF-QA | | Perception | ActivityNet-QA | |
| --- | --- | --- | --- | --- | --- | --- | --- | --- | --- |
| | Acc | Score | Acc | Score | Acc | Score | Acc | Acc | Score |
| MERV (frozen) | **70.97** | 3.76 | **59.03** | **3.25** | 51.1 | 3.26 | 46.21 | **50.87** | **3.34** |
| MERV, Video-LLaVA recipe | 70.92 | 3.78 | 58.74 | 3.25 | 51.67 | 3.27 | 47.48 | 50.42 | 3.33 |
| MERV (full) | 70.48 | **3.79** | 57.25 | 3.24 | **51.39** | **3.28** | **48.41** | 49.93 | 3.33 |
| MERV, mixed Stage 1+2 | 69.9 | 3.73 | 55.14 | 3.08 | 51.53 | 3.26 | 45.65 | 39.98 | 2.95 |

Table 5: **Ablation of training stage recipes**. We explore different training recipe strategies, starting with the standard LLaVA recipe which Video-LLaVA adopted, along with some other variations.

We also provide the full numbers for our ablations in Sections 4.2.1 and 4.2.2.

| Projector | MSVD | MSRVTT | TGIF | Perc. | Params | FLOPs |
| --- | --- | --- | --- | --- | --- | --- |
| 257 tok | 68.47 | 55.81 | 48.62 | 46.14 | - | - |
| class tok | 65.98 | 55 | 43.7 | 43.51 | - | - |
| 2D Avg | 68.23 | 56.92 | 48.99 | 45.69 | 0 | 2.1M |
| 2D Avg* | **69.08** | **58** | **50.01** | 46.34 | 0 | 4.2M |
| 2D Attn | 65.76 | 55.23 | 43.35 | 44.14 | 12.7M | 9.7G |
| 2D Conv | 67.48 | 56.78 | 47.6 | 45.04 | 237M | 241G |
| 3D Avg* | 68.62 | 57.2 | 49.59 | 44.95 | 0 | 4.2M |
| 3D Conv | 68.56 | 57.03 | 49.28 | **46.81** | 113M | 232G |

(a) **Pre-fusion projectors.** * is 16 frames instead of 8. Top two rows are projector-free baselines.

| Tkns | MSVD | MSRVTT | TGIF | Perc. |
| --- | --- | --- | --- | --- |
| 1 | 61.94 | 54.64 | 41.41 | 42.85 |
| 4 | 64.47 | 55.72 | 45.32 | 43.31 |
| 16 | 67.23 | 56.44 | 47.75 | 43.18 |
| 64 | 69.08 | 58.00 | 50.01 | 46.34 |
| 100 | 68.38 | 57.47 | 48.78 | 45.56 |
| 144 | 68.65 | 57.73 | 48.81 | 43.94 |
| 256 | 68.46 | 57.72 | 48.66 | 43.51 |

(b) **Pre-fusion output token.** We ablate the optimal token size per frame for the pre-fusion projector.

| Strategy | MSVD | MSRVTT | TGIF | Perc. | FLOPs |
| --- | --- | --- | --- | --- | --- |
| Cross-Attn | **70.97** | **59.03** | **51.1** | 46.21 | 17.19 T |
| Concat (Seq.) | 66.99 | 56.95 | 48.20 | 45.67 | 43.09 T |
| Concat (Ch.) | 70.02 | 58.08 | **51.1** | **47.36** | 16.29 T |
| Learnable W | 68.06 | 56.54 | 48.82 | 46.6 | 16.24 T |
| 25% - Mixed | 68.38 | 56.99 | 47.71 | 43.66 | 16.39 T |

(c) **Feature fusion strategy.** We compare our feature fusion strategy with concatenating the visual embeddings in either token sequence dimension or the channel dimension, learning an optimal embedding mixture weights, and training with equal 25% mixture of visual embeddings.

Table 6: **Full design choice ablation numbers.** Detailed experimental results of Tables 2a, 2b, 2c. We highlight our defaults in orange and **bold** the best results.

## B.2 SOMETHING SOMETHING V2 DETAILS

### B.2.1 SOMETHING-SOMETHING V2 - OPENENDED.

Additionally, we evaluate Something-Something V2 as an open ended QA task, where the question is "*How is the object in the video being interacted with?*", and the answer is expected to be similar to

| Methods | MSVD-QA | | MSRVTT-QA | | TGIF-QA | | Perception | ActivityNet-QA | | Avg |
|---|---|---|---|---|---|---|---|---|---|---|
| | Acc | Score | Acc | Score | Acc | Score | Acc | Acc | Score | Acc |
| All 4 encoders | **70.97** | **3.76** | **59.03** | **3.25** | **51.10** | **3.26** | 46.21 | 50.87 | **3.34** | **55.64** |
| w/o LanguageBind | 68.52 | 3.69 | 57.10 | 3.19 | 50.20 | 3.23 | 45.23 | 49.78 | 3.31 | 54.17 |
| w/o DINOv2 | 69.75 | 3.74 | 57.70 | 3.23 | 49.94 | 3.23 | 46.57 | **51.43** | **3.34** | 55.08 |
| w/o ViViT | 70.12 | 3.75 | 58.26 | 3.23 | 50.45 | 3.22 | **46.94** | 51.36 | 3.33 | 55.43 |
| w/o SigLIP | 69.85 | 3.74 | 57.55 | 3.22 | 50.27 | 3.22 | 46.20 | 50.06 | 3.32 | 54.79 |

Table 7: **Effect of Each Encoder**. Detailed results of Figure 4a

| Methods | MSVD-QA | | MSRVTT-QA | | TGIF-QA | | Perception | ActivityNet-QA | | Avg | FLOPs | Params |
|---|---|---|---|---|---|---|---|---|---|---|---|---|
| | Acc | Score | Acc | Score | Acc | Score | Acc | Acc | Score | Acc | | Overall |
| MERV | **70.97** | **3.76** | **59.03** | **3.25** | **51.10** | **3.26** | **46.21** | **50.87** | **3.34** | **55.64** | 17.19 T | 7686.0 M |
| LangBind | 68.47 | 3.71 | 55.81 | 3.16 | 48.62 | 3.19 | 46.14 | 44.72 | 3.17 | 52.75 | 41.3 T | 7147.0 M |
| DINOv2 | 65.44 | 3.62 | 53.46 | 3.09 | 41.53 | 2.96 | 42.73 | 43.39 | 3.09 | 49.31 | 40.88 T | 7046.0 M |
| ViViT | 59.95 | 3.43 | 51.81 | 3.05 | 38.1 | 2.84 | 40.2 | 43.98 | 3.16 | 46.81 | 27.12 T | 6830.0 M |
| SigLIP | 66.68 | 3.64 | 56.41 | 3.16 | 48.22 | 3.16 | 45.09 | 49.41 | 3.31 | 53.16 | 31.08 T | 6834.0 M |

Table 8: **MERV Captures Single Encoder Performances**. Detailed experimental results of Figure 4a.

| | MSRVTT-what | MSRVTT-who | MSRVTT-how | MSRVTT-when | MSVD-what | MSVD-who | TGIF-what | TGIF-how | TGIF-where |
|---|---|---|---|---|---|---|---|---|---|
| MERV | **50.62** | **77.17** | 83.96 | 72.23 | **62.68** | **84.62** | 49.44 | **53.33** | **65.34** |
| ViViT | 43.06 | 70.43 | 78.90 | 68.54 | 50.10 | 75.90 | 32.90 | 50.10 | 50.62 |
| DINOv2 | 44.54 | 73.00 | 76.95 | 65.73 | 55.71 | 82.12 | 37.14 | 50.69 | 58.40 |
| LanguageBind | 46.89 | 74.86 | 83.41 | **72.53** | 59.66 | 82.95 | 46.23 | 53.24 | 59.92 |
| SigLIP | 47.96 | 74.41 | **84.21** | 71.20 | 57.17 | 82.23 | 45.65 | 53.25 | 60.21 |

Table 9: **Performance on WH-words**. Detailed experimental results of Figure 5

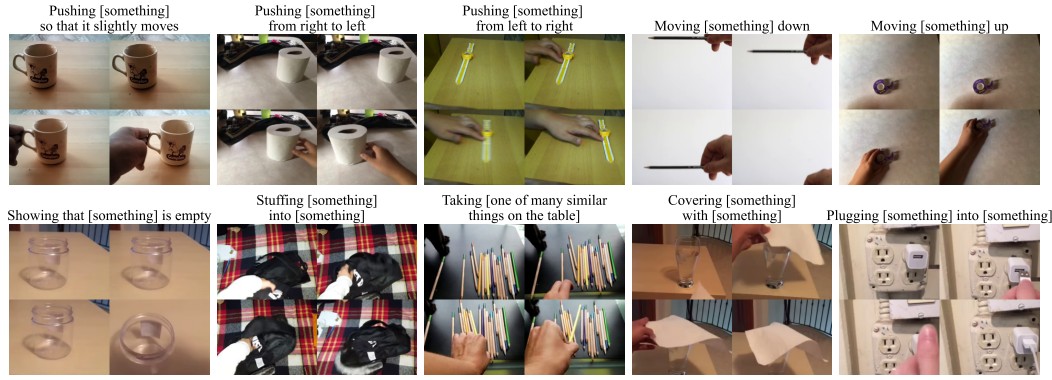

Figure 9: **Example video of Something-Something V2**. We see that ViViT show better performance in classes where temporal movement is critical for solving the task (Top row), while SigLIP performs better when the action can be inferred from the image without temporal information (Bottom row).

| | MERV | MERV-Full | LanguageBind | DinoV2 | ViViT | SigLIP | LLaMA-Vid-7B | LLaMA-Vid-13B | VideoLLaVA |
|---|---|---|---|---|---|---|---|---|---|
| Smth-Smth V2-OE-Temporal | 6.82 | **9.13** | 3.63 | 3.88 | 5.50 | 4.25 | 6.07 | 3.94 | 5.57 |
| Smth-Smth V2-OE | 17.70 | **20.65** | 13.83 | 11.03 | 10.53 | 13.84 | 16.47 | 15.62 | 19.18 |
| Smth-Smth V2-MCQ-Temporal | 36.84 | **40.65** | 30.58 | 28.08 | 39.77 | 25.39 | 27.14 | 17.89 | 22.47 |
| Smth-Smth V2-MCQ | **42.01** | 39.76 | 36.82 | 33.06 | 26.78 | 34.86 | 36.63 | 39.43 | 23.14 |

Table 10: **Performance on Something-Something V2 - OpenEnded**. These are the performance in shown Figure 4b

the class label. We use Video-ChatGPT (39)'s LLM evaluation for validating the VideoLLMs' output. Table 10 tabulates the results.

### B.2.2 SOMETHING-SOMETHING V2 - TEMPORAL.

The 12 selected classes are as following:

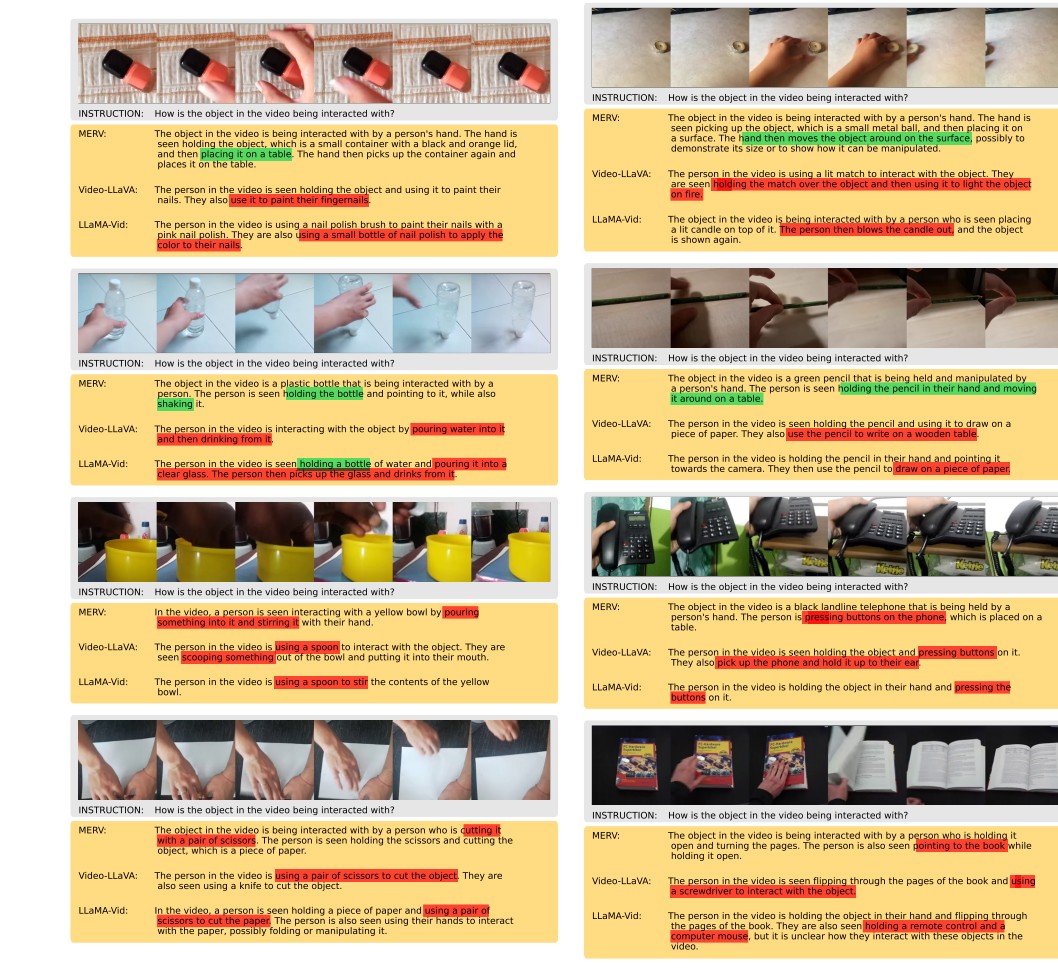

Figure 10: **Samples of MERV in SSv2.** Due to our design, our method shows better temporal action understanding than other VideoLLMs. (Top two rows) However, due to the difficulty of the task, we see failure cases for VideoLLMs. (Bottom two rows)

- Approaching [something] with your camera

- Turning the camera downwards while filming [something]

- Turning the camera left while filming [something]

- Turning the camera right while filming [something]

- Turning the camera upwards while filming [something]

- Moving away from [something] with your camera

- Moving [something] away from the camera

- Moving [something] towards the camera

- Pulling [something] from left to right

- Pulling [something] from right to left

- Pushing [something] from left to right

- Pushing [something] from right to left

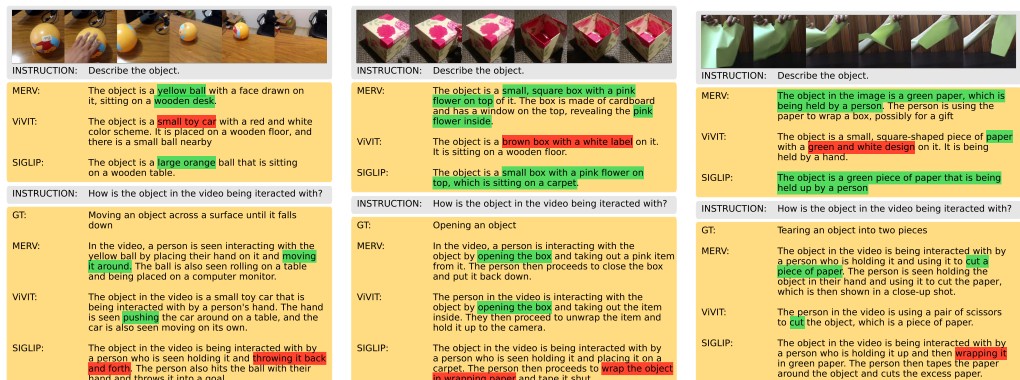

Figure 11: **Example VideoLLM output on Something-Something v2**. While SigLIP performs better on object and scene recognition, it fails to understand temporal actions. ViViT fails on the details of object recognition, but has better understanding in temporal movements.

# C ADDITIONAL ANALYSIS

## C.1 ATTENTION WEIGHTS

We look at the attention weight on our 4 benchmark datasets (MSRVTT, TGIF, MSVD, and Perception Test), and visualize the videos that have the highest attention weight for each of the encoders on Figure 12. As expected, ViViT attention weights are highest on videos with large motion, as ViViT have strong temporal motion understanding. Meanwhile, SigLIP, as they are vision-language contrastively trained, is preferred by videos that have textual data in the video. DINOv2 and Language-Bind are both preferred by videos with static scenes, but Language-Bind, as it is contrastively trained with video and language, is preferred by video with some foreground motion.

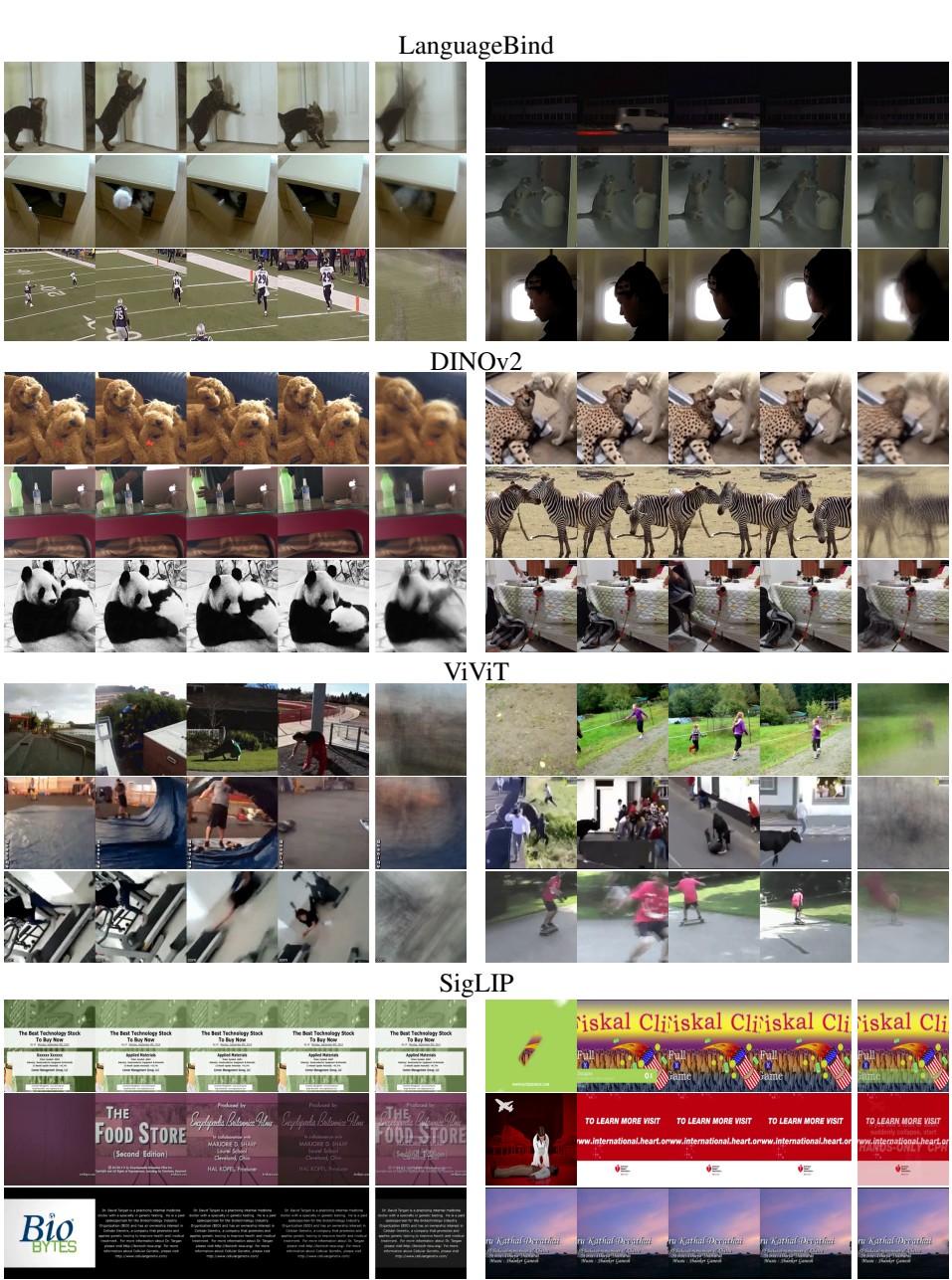

Figure 12: **Videos that give the highest attention weight for each of the encoders.** The right-most column shows the average frame of the video.

