# OpenReview forum: "Unifying Specialized Visual Encoders for Video Language Models"
_ICLR.cc/2025/Conference — Submitted to ICLR 2025_

### Official Review · Reviewer_RqfQ · 2024-10-31

**Soundness:** 3
**Presentation:** 3
**Contribution:** 3
**Rating:** 6
**Confidence:** 4

**Summary:**

Video Large Language Models (VideoLLMs) currently use only a single vision encoder while different vision encoders have their own strengths. This leads to limited capabilities of resulting VideoLLMs on various downstream tasks. The paper proposes a well-established framework for training VideoLLMs using multiple vision encoders. It empirically ablates the following four aspects: i) which visual encoders to use, ii) how to align and fuse visual features from different vision encoders, and iii) training recipes and data mixture. The authors also implement an efficient feature extraction and projection pipeline for efficiently using multiple vision encoders.

**Strengths:**

This paper is one of the pioneering work that establish the training framework for using multiple vision encoders for VideoLLMs
- by demonstrating the model's superiority on multiple downstream tasks
- through a bunch of ablation studies for validating the design choices and answering the key questions

Although the paper does not propose novel methodologies, I think it is quite valuable because there are a flood of VideoLLMs that are trained using their own data mixture and recipes. Researchers are quite confused and have questions like which vision encoder to use, whether to pretrain the adapter between the vision encoder and LLM, whether to unfreeze the vision encoder or LLM in Stage 1 training, etc. The paper provides extensive empirical results answering these questions especially for VideoLLMs using multiple vision encoders.

**Weaknesses:**

Overall the paper is quite good but answering the following concerns would make the paper better and more self-contained:

- After finishing reading, I am still not sure when I should use the MERV training recipe and when to use the MERV-Full. It would be great to give some general instructions on this.
- The chosen video model, ViViT, was published in 2021. It would be great to provide the empirical results with other recent methods including both supervised and unsupervised (self-supervised), e.g., Video Swin Transformer [1], MViTv2 [2] for supervised and VideoMAE v2 [3] for unsupervised.
- Figure 4 (b) and Figure 5 illustrate and disentangle the individual contributions of contrastive methods and the video model, ViViT, to model performance, but I cannot figure out whether there are distinct differences between contrastive methods from the figures. I cannot even find noticeable trends in Figure 3, e.g., DINOv2 outperforms SigLIP on MSRVTT-Who while it lags behind SigLIP on MSVD-Who.
- In Section 3.4, it would be great to elaborate on how to make feature extraction and projection happen in parallel.
- The authors often claimed in the paper that they found video-language alignment was not very strong for the MERV recipe, e.g., L393-L394. How did the authors observe or verify that?

[1] Ze Liu et al., Video Swin Transformer, CVPR 2022.
[2] Yanghao Li et al., MViTv2: Improved Multiscale Vision Transformers for Classification and Detection, CVPR 2022.
[3] Limin Wang et al., VideoMAE V2: Scaling Video Masked Autoencoders with Dual Masking, CVPR 2023.

**Questions:**

Please address the above concerns, and there are some minor questions:

- Are single encoder models in Figure the variants of MERV?
- What is the role of P_e: to differentiate between different encoder features? If then, why did the authors name it "positional" embeddings? And why does MERV need this explicit differentiation between the encoder features?
- L254-L255: Does it take 24 hours with 8 L40-48GB GPUs for MERV or MERV-Full?
- Table 1: Please add the columns to specify which vision encoder and llm are used for each method, and TGIF score for Video-ChatGPT is missing in Table 1 (3.0, L309)
- L338: Does 3D Conv in Table 2 (a) apply a 2D 3x3 convolution? Naming is confusing.

Also, please fix typos in the paper.

---

> ### Author Response · Authors · 2024-11-22
> **Response (1/2)**
>
> Dear Reviewer RqfQ, we deeply appreciate your valuable reviews. We are encouraged that you appreciate our model's performance that outperforms existing baselines on different benchmarks, extensive ablation studies, and empirical experiments and results that can answer the questions in a controlled and scientific manner. Please see our responses to the weaknesses you pointed out below:
>
> **W1: MERV or MERV-full.**
> Given that MERV (full) is trained on more data, we believe that it will offer better generalization. We have revised the manuscript to make this recommendation (line 299).
>
> **W2: ViViT is an outdated video encoder.**
> Thank you for the suggestion. Given limited timeline, we conducted experiments using one newer SoTA video model Hiera [1], i.e. MViTv3, in a single-encoder setting, four-encoder setting replacing ViViT, and a five-encoder setting. The checkpoint we use was pretrained using the Masked Autoencoder (MAE) self-supervised learning technique on Kinetics 400 (K400), and then finetuned on K400 using supervised learning. On the other hand, ViViT was initialized from an ImageNet-21K/JFT backbone, and trained on K400 using supervised learning, so their training setups are similar.
>
> The results are presented in the table below, with the best result in bold and the second-best in italics. Replacing ViViT with Hiera demonstrates improvements on the fine-grained spatiotemporal reasoning benchmark, Perception Test, with accuracy gain of 1.29%. Similarly, adding Hiera yields an improvement on ActivityNet, achieving a 0.36% increase in accuracy. However, on other benchmarks, the original MERV remains the strongest model. Overall, we observe no significant performance improvement when training with Hiera, which aligns with expectations, since Hiera is under the same paradigm as ViViT, functioning as a temporal expert trained on short videos. We also hypothesize that Hiera is more sensitive to the temporal stride than ViViT, as ViViT can reasonably deduce motion from uniformly sampled frames. We expect performance to improve if we incorporate encoders trained on different paradigms and data sources or process a much greater number of frames simultaneously, which we will leave for future work.
>
> | Method | MSVD -Acc | MSVD -Score | MSRVTT -Acc | MSRVTT -Score | TGIF -Acc | TGIF -Score | ActivityNet -Acc | ActivityNet -Score | Perception Test -Acc |
>  | --- | --- | --- | --- | --- | --- | --- | --- | --- | --- |
>   | VideoLLaVA | 67.74 | 3.69 | 56.90 | 3.18 | 47.99 | 3.17 | 47.08 | 3.27 | 44.22 |
>  | MERV | **70.97** | **3.76** | **59.03** | **3.25** | **51.1** | **3.26** |  *50.87* | **3.34** | 46.21 |
> | MERV,  ViViT replaced with  Hiera | *69.68* | *3.74* | 57.64 | 3.22 | *50.38* | *3.24* | 50.24 | **3.34** | **47.50** |
>  | MERV + Hiera | 69.67 | 3.72 | *58.26* | *3.23* | 50.32 | 3.22 | **51.23** | **3.34** | *46.23* |
> | ViViT Single Encoder LLM | 59.95 | 3.43 | 51.81 | 3.05 | 38.10 | 2.84 | 43.98 | 3.16 | 40.20 |
> | Hiera Single Encoder LLM | 55.38 | 3.28 | 49.21 | 2.95 | 36.02 | 2.76 | 44.01 | 3.15 | 40.20 |
>
> [1] Ryali et. al., Hiera: A Hierarchical Vision Transformer without the Bells-and-Whistles, ICML 2023.
>
> **W3: Distinctive Differences among Contrastive Models.**
> We have included qualitative analysis for each of the visual encoders in the appendix. Figure 12 visualizes videos that give the highest attention weight for each of the encoders. By looking at which videos that the feature fusion module assigns high attention weight, we can see what type of videos that each encoder specialize on.
> We see that SigLIP is preferred when given text-heavy videos, while ViViT is preferred when the video contains a lot of temporal motion. Both LanguageBind and DINOv2 are preferred with static videos, but LanguageBind seems to be more preferred when there are meaningful foreground motions.
>
> **W4: Elaborate on how to make feature extraction and projection happen in parallel.**
> Thank you for the feedback, we’ve discussed our use of FSDP in greater detail in Section 4.2.3 under Training Recipes now. Simply put, we use the default FSDP sharding strategy in PyTorch as it is not currently possible to specify an explicit plan for which modules go where (but this may be possible as FSDP matures). However, we design our code so that no operations block each other in this step, so that FSDP can find an optimal sharding strategy automatically (which we believe it does, as shown by our small overhead in the new Figure 3).

---

> ### Author Response · Authors · 2024-11-22
> **Response (2/2)**
>
> **W5: How do you verify that the video-language alignment was not very strong for the MERV (frozen) recipe?**
> MERV (frozen) does not use stage 1 data, which largely consists of captioning and video description tasks focused on visual alignment. Although our testing benchmarks don’t reveal a stark difference in the results, we do see some performance hits on other benchmarks. In our previous explorations, the distributions of language used in video datasets and benchmarks also seem to sparsely overlap based on their sentence embeddings, which could have impacted our ability to perform well zero-shot on the downstream benchmarks (especially difficult, more OOD ones). Therefore, we make this observation mainly based on the difference in data used to train MERV (frozen) and MERV (full), as our assumption is that more data will help further with this alignment gap. We add this clarification in the revised manuscript (lines 387-390).
>
> **Q1: Are single encoder models in Figure the variants of MERV?**
> In all of the Figures, any single-encoder models are our MERV architecture and recipe except with a single encoder and no feature fusion. The only exception is in Figure 4a, where we plot the models with their full embeddings to also demonstrate the impact of our pre-fusion projectors on performance and efficiency (line 422).
>
> **Q2: What is the role of P_e?**
> Role of P_e is to differentiate between different encoders, where we call it positional embedding for each encoder following the similar notation used in other attention models. We have added them so that the feature fusion module has additional knowledge on which key belongs to which encoder, but we have seen that P_e does not have a strong impact on the performance. Thank you for pointing out, we decided to remove this for clarity.
>
> **Q3: Does it take 24 hours with 8 L40-48GB GPUs for MERV (frozen) or MERV (full)?**
> It takes around 24 hours on MERV (frozen), while MERV (full) takes 56 hours as they are trained using 2 stages with a larger dataset.
>
> **Q4: Table 1. Add columns to specify video encoder and LLM used. TGIF missing for Video-ChatGPT.**
> Thank you for the suggestion, we added the video encoder and LLM used by the model in Table 1 (also in Table 3 and Table 4). Regarding TGIF performance for Video-ChatGPT, we apologize for the confusion. We decided to remove TGIF performance from Video-ChatGPT as they were evaluated on the FrameQA subset of the TGIF instead of the full test set (https://github.com/PKU-YuanGroup/Video-LLaVA/issues/37#issuecomment-1900058137), but did not update the main text correspondingly. We also removed the point about score (line 308 on the previous draft), as now the accuracy and score trends more consistently.
>
> **Q5: Does 3D Conv in Table 2 (a) apply a 2D 3x3 convolution?**
> Thanks for catching this typo, it is a 3D Conv, i.e. 2x3x3 (FxHxW). We have added additional clarifications for this in the beginning of Section 4.2.1 on our pre-fusion projector ablations.

---

> > ### Comment · Reviewer_RqfQ · 2024-11-25
> >
> > Thank the authors for the thorough response, including additional experiments. I will keep my rating unless discussions with other reviewers change my mind.

---

> > > ### Author Response · Authors · 2024-11-29
> > > **Thank you for your response!**
> > >
> > > Dear Reviewer RqfQ,
> > >
> > > Thank you very much for your prompt response and the valuable time you’ve dedicated to reviewing our work and reading our replies! Since there is still time remaining in the discussion period, we would be more than happy to address any additional questions or suggestions you may have to further improve the quality of the paper. Thank you once again for your time and thoughtful feedback!
> > >
> > > Sincerely,
> > > Submission #545 Authors

---

### Official Review · Reviewer_ccLB · 2024-11-04

**Soundness:** 3
**Presentation:** 2
**Contribution:** 2
**Rating:** 5
**Confidence:** 4

**Summary:**

This paper proposes using multiple specialized visual encoders as ensembles to enhance the representation of videos in visual language models. The goal is to improve the model's ability to capture diverse visual information by leveraging the strengths of different encoders. The authors incorporate visual encoders such as SigLIP, DinoV2, ViViT, and LanguageBind, each contributing unique spatial, temporal, and multimodal understanding capabilities, making the ensemble more comprehensive. A feature fusion module is introduced to combine features from these visual encoders. The fusion process involves a cross-attentive encoder mixer, which aligns and integrates features from different encoders, allowing for a unified representation that retains important spatio-temporal details. The authors evaluated different combinations of visual encoders using the proposed fusion module on multiple video understanding benchmarks, including MSVD-QA, ActivityNet-QA, and Something-Something v2. Improvements in accuracy were observed, demonstrating the potential of the multi-encoder approach.

**Strengths:**

1. **Multi-Encoder Approach:** The use of multiple specialized visual encoders aims to provide a more comprehensive understanding of video content by leveraging the capabilities of each encoder. This approach attempts to capture diverse types of visual information, including spatial, temporal, and multimodal aspects, which may enhance the model's performance on video language tasks.

2. **Experimental Design:** The authors provide an extensive experimental study of the combination of visual encoders, considering individual and joint usage. This exploration offers insights into practitioners into the effectiveness of different encoder combinations. Additionally, the authors analyze the impact of training stages in LLaVa-style VLM training for videos, providing observations on the importance of each stage in the two-stage training process.

3. **Performance on Benchmarks:** The proposed method demonstrates improvements in video understanding accuracy across multiple tested benchmarks. The reported accuracies outperform existing baselines, suggesting the potential benefits of the multi-encoder fusion strategy.

4. **Efficiency and Implementation:** The proposed fusion module is designed to be lightweight and straightforward to implement, which makes it accessible for integration into existing systems. This ease of implementation could facilitate adoption without requiring substantial computational resources or complex modifications.

**Weaknesses:**

1. **Novelty Concerns:** The use of multiple visual encoders has become a common paradigm for visual language models [R1, R2, R3], primarily for image-based visual input. The authors have not provided sufficient reasoning to demonstrate the novelty of extending this approach to video. The method mainly involves passing multiple video frames through image-based visual encoders, with ViViT being the only video-specific embedding model. This limits the originality of the proposed multi-encoder approach.

2. **Feature Fusion Module Effectiveness:** The proposed feature fusion module lacks a clear advantage. Table 2 indicates that the cross-attention feature fusion yields nearly the same accuracy as a simple channel concatenation approach. Additional statistical significance testing might be needed to substantiate the performance difference if it is considered significant by the authors. Without stronger evidence, the proposed fusion module may not provide enough of a meaningful contribution.

References:

[R1] Tong et al. Cambrian-1: A Fully Open, Vision-Centric Exploration of Multimodal LLMs, arXiv:2406.16860, 2024

[R2] Lin et al. SPHINX: The Joint Mixing of Weights, Tasks, and Visual Embeddings for Multi-modal Large Language Model. arXiv:2311.07575, 2023

[R3] Jain et al. VCoder: Versatile Vision Encoders for Multimodal Large Language Models. arXiv:2312.14233, 2023

**Questions:**

1. **Computational Trade-offs:** Given that using multiple visual encoders improves accuracy at the cost of increased computational demands, can the authors provide a detailed analysis of the runtime differences between using a single encoder versus multiple encoders?

2. **Fusion Module Significance:** Can the authors clarify the significance of the proposed feature fusion module in comparison to simpler methods? What specific scenarios or tasks benefit most from this module compared to other straightforward techniques like concatenation?

---

> ### Author Response · Authors · 2024-11-22
> **Response (1/2)**
>
> Dear Reviewer ccLB, We deeply appreciate the time and the effort that you have shown on the review. Specifically, we thank you for appreciating our multi-encoder approach, extensive experimental studies for different encoder combinations and training stages, improvements to existing baselines, and lightweight, resource-friendly design. Please see our responses to the weaknesses you pointed out below:
>
> **W1: The authors have not provided sufficient reasoning to demonstrate the novelty of extending this approach to video.**
>
> We agree that we are not the only ones working in this space of multiple visual encoders, which has very recently become popular. We want to re-emphasize that we have two video models in our mixture, as both ViViT and LanguageBind are trained on and embed videos. Additionally, while our method seems simple, we performed extensive ablations to find that such simple approaches are both enough and necessary for flexibly and efficiently modeling *videos* using multiple image and video encoders (Section 4.2, Table 2). We conducted a very in-depth analysis of how the behaviors of these image-based models differ from video-native models when processing images (Section 5, Figure 4,5,6,12). Prior VideoLLMs largely rely on image models for visual understanding, a paradigm we aimed to challenge with our work. We showed concrete evidence of how useful video models can be for temporally sensitive tasks such as on Something-Something v2 (Figure 4.b, Figure 6), and we added additional analysis per other reviewer’s requests for patterns to how the feature fusion chooses encoders (see the below discussion in **W2**). Some of our discussion is enabled by our specific feature fusion design, which departs from other conventions in VideoLLM design (e.g., line 346 to line 350).
>
> However, it is also not straightforward to simply incorporate any video encoder for this to work. As requested by other reviewers, we performed new ablations on using extra video encoders, and found that using a new video encoder with similar knowledge areas to our pre-existing encoders does not help much. For example, adding or substituting ViViT with Hiera [1], a short-video encoder similar to ViViT, results in comparable but overall slightly worse gains over VideoLLaVA than our current MERV (Results are shown below, with the best in **bold** and the second-best in *italics*).
>
> | Method | MSVD -Acc | MSVD -Score | MSRVTT -Acc | MSRVTT -Score | TGIF -Acc | TGIF -Score | ActivityNet -Acc | ActivityNet -Score | Perception Test -Acc |
>  | --- | --- | --- | --- | --- | --- | --- | --- | --- | --- |
>   | VideoLLaVA | 67.74 | 3.69 | 56.90 | 3.18 | 47.99 | 3.17 | 47.08 | 3.27 | 44.22 |
>  | MERV | **70.97** | **3.76** | **59.03** | **3.25** | **51.1** | **3.26** |  *50.87* | **3.34** | 46.21 |
> | MERV, ViViT replaced with Hiera | *69.68* | *3.74* | 57.64 | 3.22 | *50.38* | *3.24* | 50.24 | **3.34** | **47.50** |
>  | MERV + Hiera | 69.67 | 3.72 | *58.26* | *3.23* | 50.32 | 3.22 | **51.23** | **3.34** | *46.23* |
>
> [1] Ryali et. al., Hiera: A Hierarchical Vision Transformer without the Bells-and-Whistles, ICML 2023.
>
> **W2: The proposed feature fusion module lacks a clear advantage. Additional statistical significance testing should be considered.**
>
> Thank you for pointing that out. We agree that it is important to show that our method performs better than the concatenation method. We re-ran our method and the concat method using two more random seeds. The mean accuracy and standard deviation are shown in the table below. We find that our method is still better than the concat method on average. We ran a paired t-test which returned $p=0.063$; thus, the difference is trending towards significance, but due to compute limitations and the limited timeline we could only do 3 seeds (we’re working on more). Another additional benefit of cross-attention is having accessible encoder weightings for analysis, so we do not choose channel-wise concatenation as our final design.
>
> To this point, the cross-attention method provides explicit attention weights that we can look at to know which videos the feature fusion tends to weigh the most, and look for trends among those. In this setting, we checked the attention weights of all the videos in 4 of our benchmark datasets (MSRVTT, MSVD, TGIF, and Perception Test). For each encoder, we visualized the videos which give the highest attention weight for that specific encoder. The qualitative results are added in the appendix of the paper (Figure 12).  As a summary, we found that ViViT attends to videos with large motion, whereas SigLIP is more towards videos with text data (and often more static videos). Meanwhile DINOv2 and LanguageBind have a bias towards videos with static scenes, but LanguageBind attends more often to videos with foreground motion thanks to its video pretraining. This also demonstrates the benefit of using a cross-attention feature fusion, which allows us to analyze the model in this way.

---

> ### Author Response · Authors · 2024-11-22
> **Response (2/2)**
>
> | Feature Fusion Strat. | Average Accuracy | MSVD | MSRVTT | TGIF | Perception Test |
> | --- | --- | --- | --- | --- | --- |
> | MERV, Cross-Attention | **56.66** ± 0.51 | **70.54** ± 0.45 | **58.45** ± 0.52 | **50.66** ± 0.41 | **46.98** ± 0.67 |
> | MERV, Channel Concat | 56.25 ± 0.52 | 70.15 ± 0.26 | 57.95 ± 0.21 | 50.65 ± 0.81 | 46.42 ± 0.82 |
>
> **Q1: Runtime difference between single encoder vs multiple encoders?**
>
> The following table shows the runtime of individual encoder models:
> | Single Encoder | Step Time (s) |
> | --- | --- |
> | DINOv2 | 13.60 |
> | LanguageBind | 14.04 |
> | SigLIP | 13.28 |
> | ViViT | 14.47 |
>
> We provide the detailed runtime numbers on 8xL40-48GB GPUs in the tables here and also in our revised manuscript in Figure 3. We track the step time as we progressively add more encoders in a multi-encoder architecture, with the ordering of DINOv2, LanguageBind, SigLIP, and ViViT. We also compare each multi-encoder model (top row in the table below) with an equivalent single encoder model which has the slowest time (bottom row in the table below). For example, we compare DINOv2+LanguageBind with the time of the LanguageBind-only model, as this provides the most equitable comparison of what a theoretical best runtime would be with perfect parallelization (e.g. hiding the DINOv2 encoder runtime entirely under the LanguageBind). Note that this theoretical best runtime doesn’t take into account the overhead of the feature fusion, which remains the largest source of incurred cost remaining.
>
> | Number of Encoders  | 1 | 2 | 3 | 4 |
> | --- | --- | --- | --- | --- |
> | Avg. step time using first N-encoders | 13.60 ± 0.38 | 14.82 ± 0.28 | 14.95 ± 0.31 | 16.31 ± 0.78 |
> | Avg. step time using the slowest single encoder among first N-encoders | 13.60 | 14.04 | 14.04 | 14.47 |
>
> Comparing the runtime numbers in the two tables, adding multiple encoders only marginally increases the runtime, thanks to our efficient implementation and a feature fusion strategy that does not increase the token length. (The 1 stdev variance is variable of the machine, which tends to happen even after a few iterations of warmup.)
>
> **Q2: Fusion Module Significance. What scenarios does our method perform better than the naive techniques?**
>
> We have tested different feature fusion strategies and tabulated their performance in Table 2.c and Table 6.c in the appendix. We find our methods to be better performing in overall performance (17.19 T Flops, accuracy 56.83), while having less computation than sequence-wise concatenation (43.09 T Flops, accuracy 54.45), and on-par with channel-wise concatenation (16.29 T Flops, accuracy 56.64). Additionally, our feature fusion module performed better or on-par with channel-wise concatenation in 3 out of 4 benchmarks that we have tested (Table 6.c), having robust performance towards unseen datasets.
>
> Moreover, similar to Figure 5 (Figure 3 in the original submission), we have looked at the performance difference on WH-words of three open-ended benchmarks.
>
>  | Feature Fusion Strat. | MSRVTT-what | MSRVTT-who | MSRVTT-how | MSRVTT-when | MSVD-what | MSVD-who | TGIF-what | TGIF-how | TGIF-where |
>  | --- | --- | --- | --- | --- | --- | --- | --- | --- | --- |
>  | Channel Concat | 49.87 | 75.89 | 82.50 | 69.38 | 62.16 | 83.23 | 49.39 | 53.31 | **66.19** |
>  | Cross Attention (ours) | **50.62** | **77.17** | **83.96** | **72.23** | **62.68** | **84.62** | **49.44** | **53.33** | 65.34 |
>
> This shows that our method performs consistently better over all but one the visual tasks of these three benchmarks.
>
> We hypothesize that channel-wise concatenation may be preferable to cross-attention in test scenarios that differ significantly from training and demand intensive, fine-grained spatiotemporal reasoning. This is because concatenation naively retains all information with equal importance, whereas cross-attention selectively learns to prioritize information through soft attention, which may fail to generalize effectively when trained on limited training datasets.

---

> > ### Comment · Reviewer_ccLB · 2024-11-27
> >
> > Thank you for responding to the comments. The standard deviation of accuracy between cross-attention and channel concat indicates that the improvement is marginal. For the argument that "another additional benefit of cross-attention is having accessible encoder weightings for analysis", channel concat can also be analyzed by probing the linear embedding following the concatenated feature. I don't have additional experiments to request and keep my rating.

---

> > > ### Author Response · Authors · 2024-11-30
> > > **Thank you for your response!**
> > >
> > > Dear Reviewer ccLB,
> > >
> > > Thank you for your prompt response! We appreciate your directed response to parts of our rebuttal. We hope that our reply regarding the model novelty was satisfactory for you.
> > >
> > > Regarding the significance of the improvements provided by our cross-attention-based feature fusion over channel-wise concatenation, we would like to clarify that a paired t-test is a more appropriate metric for significance testing in this context than standard deviation. The paired t-test assesses whether the observed differences between means are likely due to random chance, whereas standard deviation does not compare groups or test for significance. Instead, it measures the variation or dispersion of individual data points of a single group.
> > >
> > > Our paired t-test yielded a p-value of 0.063 based on three seeds across four benchmarks. While slightly above the conventional threshold of 0.05, this p-value suggests a trend toward significance, especially given the limited number of random seeds with which we were able to run experiments and the consistent improvements observed across all datasets. We will show results with more seeds; we hope to show results with more seeds, but due to the high cost of training and evaluation that requires GPT-3.5 queries, we are only able to show results of three seeds currently. With three random seeds, cross-attention demonstrates consistent improvements over channel-wise concatenation across the four benchmarks we tested, with an average accuracy gain of 0.41%. While this improvement is modest, it is meaningful in light of the p-value and the consistent results, which underscore the robustness of our method. We hope that demonstrating consistent performance improvements across all four benchmarks, alongside the observed p-value, will convince you of the statistical significance of our improvements.
> > >
> > > With regard to the interpretability of cross-attention compared to channel concatenation, we agree that in theory one could probe the linear embeddings following the concatenated features to attempt similar analyses. However, the point of cross-attention is that it does the probing of the linear embedding already, so it’s a best of both worlds that we can achieve this with our method. If we were to probe the entire channel concatenated output with the naive standard linear layer probe, different weights would be given to different channel layers, making it difficult to isolate encoder-specific contributions and perform a clean, in-depth analysis on the feature fusion. One could average across all of the weights of one encoder’s channels, but we believe this is not an accurate analysis method. Aggregating or grouping these weights for analysis would effectively replicate the behavior of cross-attention, which already achieves this mixing in a principled and interpretable manner. In contrast, cross-attention directly integrates encoder representations in a structured way that inherently facilitates interpretability and analysis, avoiding the need for additional probing or post-hoc processing.
> > >
> > > Finally, we also want to highlight the broader contributions of our work beyond feature fusion. While we were not able to convince you with one of our ablation studies in the feature fusion design, we want to emphasize that our work encompasses multiple more contributions, including selection and validation of multiple vision and video encoders with different visual skills, pre-fusion projector designs for spatio-temporal alignment of visual embeddings, and out-performing SOTA methods while using same or less training data with in-depth analyses. While the feature fusion design may represent only one component of our system, we believe our choice is well-motivated, efficient, and contributes to a more condensed and interpretable representation. Together, these elements provide a holistic contribution that we believe advances the field.
> > >
> > > We deeply value your fruitful insights and comments! Should there be any remaining concerns, we would be happy to address them further. We appreciate the opportunity to address your concerns. Otherwise, we sincerely hope you might consider raising your score in light of our clarifications and the contributions of our work. Thank you once again for your constructive feedback and time!
> > >
> > > Sincerely,
> > > Submission #545 Authors

---

### Official Review · Reviewer_6nZ3 · 2024-11-04

**Soundness:** 3
**Presentation:** 3
**Contribution:** 2
**Rating:** 5
**Confidence:** 5

**Summary:**

The paper propose a encoder ensembing method to mitigate the shortcoming of a single visual encoder for video understanding. The fusion module is based on cross-attention layers. Extensive experiments are conducted on multimodal benchmarks like MSRVTT, TGIF, and motion-oriented visual benchmark, SSV2. Quantitatively and qualitatively reulsts verifies the skill specializations of different visual experts and better performance is achieved compared with state-of-the-art methods.

**Strengths:**

- The motivation is reasonable. As videos contain both static and dynamic cues across diverse objects and scenes, different video encoders may capture different parts of the video and could help each other.

- The presentation is mostly clear.

- Technical details are clearly described and the reproducibility is good.

**Weaknesses:**

- While a performance boost is observed, my primary concern is that the novelty of this paper is limited. Similar to [1], the paper presents an empirical solution for ensembling multiple visual encoders in multimodal models. The feature fusion operation relies on cross-attention and linear projection layers, lacking an in-depth analysis of feature interactions. As ensembling is the basic idea in machine learning, this paper does not provide new insights, and the performance improvement is unsurprising given the increased computational costs.

[1] Liu et al., Prismer: A Vision-Language Model with Multi-Task Experts, 2023.

- Scalability of the proposed method is unclear. According to the experiments in Table 4, it appears that the only video backbone, ViViT, has a slight influence on the final performance. Combined with the results in Figure 5, there seems to be a contradiction regarding the effectiveness of a temporal-oriented feature encoder when applied to videoMLLM benchmarks. In light of this, I am concerned about the extensibility of the proposed method. To what extent can the model be applied to additional visual backbones?

**Questions:**

Please see weaknesses.

---

> ### Author Response · Authors · 2024-11-22
> **Response (1/3)**
>
> Dear Reviewer,
>
> Thank you for giving us your time and energy for the thorough and insightful review of our paper. We are encouraged that you find our motivation reasonable, appreciate our technical details and reproducibility needed to run our many experiments, and the presentation of method and results clear. Please see our responses to the weaknesses you pointed out below:
>
> **W1: Lack of novelty for three reasons: a) Similarity to Prismer for ensembling (which is a basic idea), b) Lack of in-depth analysis of feature fusion interactions, and c) Unsurprising performance given increased computational costs.**
>
> Thank you for this comprehensive set of points, and we agree that these are important points to address for our method. We will discuss them in detail one by one.
>
> a) We appreciate the reviewer for sharing the Prismer paper; we add it to our related work for relevance to our work (line 142). However, there are a few differences we think are pertinent. First, we would classify Prismer as a multimodal work, i.e. it uses segmentation, depth, normals, etc. as input, whereas ours strictly uses RGB as visual input, which we attempt to quantify as “multi-encoder”. We explore how different objectives and datasets create different visual experts, rather than the task itself. Second, Prismer doesn’t handle video, and their Experts Resampler is effectively a Perceiver Resampler for reducing the multimodal input to a fixed token length. We actually tested this type of approach (line 316), but we found that such approaches which are agnostic to structure lead them to being weaker compared to other alternatives (line 348). This reinforces our belief that a structure-aware spatio-temporal alignment is crucial for multi-encoder models to perform well to deal with videos (see point (c)). Finally, while we find Prismer a pioneering work in this aspect, we do acknowledge that they found great difficulty in improving the performance with more experts. Even with four additional experts, their VQAv2 accuracy improves from 72.17% to 72.79%, a gain of 0.62%, whereas we see average accuracy increases of 2.5% on top of the strongest single-encoder methods. We believe that these points are enough to significantly differentiate our methods and findings from Prismer.
>
> b) With regard to analysis of feature fusion interactions, we found that the model is finding the best weights to perform a general unification of the features, rather than learning large changes in weights for different classes of examples to perform single or dual encoder selection. This is understandable from the objective, but also that individual videos can give rise to many different tasks requiring different skills, and *a priori* of the task, the model simply tries to learn the best representation for all potential questions.
>
> Nevertheless, we can still look at specific videos for which the feature fusion module tends to rely more heavily on particular encoders, and look for trends among those. In this setting, we checked the attention weights of all of the videos in 4 of our benchmark datasets (MSRVTT, MSVD, TGIF, and Perception Test). For each encoder, we visualized the videos which give the highest attention weight for that specific encoder. The qualitative results are added in the appendix of the paper (Figure 12). As expected, ViViT attention weights are highest on videos with large motion, as ViViT has strong temporal motion understanding. Meanwhile, SigLIP, which is vision-language contrastively trained, attends more to videos that have textual data in the video. DINOv2 and LanguageBind are both preferred by videos with static scenes, but LanguageBind, being trained on videos, is preferred by videos with some foreground motion.

---

> ### Author Response · Authors · 2024-11-22
> **Response (2/3)**
>
> c) Finally, we respectfully disagree that increased computational costs directly lead to performance improvements. We provide three points of evidence from our ablations. First, in Table 2(a), we provide a total of six 8-frame projector settings, the first of which (257 tok) is equivalent to the original VideoLLaVA. Three of the remaining five settings led to worse performance, despite using more params and FLOPs than our 2D average, which was unintuitive for us as well given their strong performance for other domains. In fact, the most computationally demanding projectors like a 2D Conv did not perform as well as the zero parameter 2D Average Pool. A similar finding which is a classic issue with video models is ablating the number of tokens used, as using multiple frames quickly exhausts the model’s capacity. We found that 64 tokens was the best performing setting, which was counter to what previous works such as Video-LLaVA did with full token embeddings. Finally, we also performed a new set of ablations on using extra video encoders, and found that using a new video encoder with similar knowledge areas to our pre-existing encoders does not help much. For example, adding or substituting ViViT with Hiera [1], a short-video encoder similar to ViViT, results in comparable but overall slightly worse gains over VideoLLaVA than our current MERV, indicating that adding or choosing encoders naively does not improve performance as effectively as our current approach (Results are shown below, with the best in **bold** and the second-best in *italics*.).
>
> | Method | MSVD -Acc | MSVD -Score | MSRVTT -Acc | MSRVTT -Score | TGIF -Acc | TGIF -Score | ActivityNet -Acc | ActivityNet -Score | Perception Test -Acc |
>  | --- | --- | --- | --- | --- | --- | --- | --- | --- | --- |
>   | VideoLLaVA | 67.74 | 3.69 | 56.90 | 3.18 | 47.99 | 3.17 | 47.08 | 3.27 | 44.22 |
>  | MERV | **70.97** | **3.76** | **59.03** | **3.25** | **51.1** | **3.26** |  *50.87* | **3.34** | 46.21 |
> | MERV,  ViViT replaced with  Hiera | *69.68* | *3.74* | 57.64 | 3.22 | *50.38* | *3.24* | 50.24 | **3.34** | **47.50** |
>  | MERV + Hiera | 69.67 | 3.72 | *58.26* | *3.23* | 50.32 | 3.22 | **51.23** | **3.34** | *46.23* |
>
> [1] Ryali et. al., Hiera: A Hierarchical Vision Transformer without the Bells-and-Whistles, ICML 2023.
>
> As reviewer RqfQ has said, there are a flood of VideoLLMs that are trained using their own data mixture and recipes, which makes our paper particularly valuable as we made non-negligible efforts in providing fair comparisons and extensive empirical results answering various questions especially for VideoLLMs using multiple vision encoders. Ours is the first work that establishes the training framework for using multiple encoders for VideoLLM, and we found that some methods which work for VLMs such as the C-Abstractor or even other VideoLLMs like Perceiver Resamplers do not work well for multi-encoder representation learning of videos.
>
> We hope that our attention to detail and extensive analysis to the specific behaviors of our model sets our method apart from the others, as we believe that these types of details matter quite a lot. We believe current video LLM works could provide deeper analysis of their methods and differences, so we went to great lengths to make these comparisons as fair and general as possible so that the community can find broad insights.

---

> ### Author Response · Authors · 2024-11-22
> **Response (3/3)**
>
> **W2: The only video backbone, ViViT, has a slight influence on the performance while it performs well in SSv2. Unclear extensibility of the proposed method to additional visual backbones.**
>
> To answer the first question, we have included two video-encoders, ViViT and LanguageBind, in the original submission. While LanguageBind shows a decent influence on the final performance, ViViT seems to be lacking, as shown in Table 7 (Table 5 in the original PDF). We hypothesize that since ViViT is pre-trained only with videos on a classification task, it might relatively lack the vision-language semantic understanding required for the Video-QA tasks in Table 7. However, as seen in Figure 4b, ViViT demonstrates superior performance on tasks where temporal sensitivity and understanding are critical, thus enhancing MERV's performance on such tasks. While the contribution of ViViT may appear small on some of the Video-QA benchmarks we tested, this may also be due to the relative scarcity of the previously mentioned tasks in these benchmarks (e.g., researchers have found that ImageLLMs, though lacking any explicit temporal modeling or training, could have high performance on benchmarks like MSVD-QA [2][3]). We do not believe that ViViT has a slight influence but rather that it plays a very important role, as it is specialized in skills that other encoders lack.
>
> As for the second question, we found that the additional visual encoder is only helpful if the newly added encoder brings visual expertise that is not covered by the existing choice of encoders. We conducted new experiments using another SoTA video model Hiera-B+ [1], i.e. MViTv3, under the same visual expertise as ViViT (see Table above in W1), specialized in temporal understanding, both trained with Kinetics-400. We see that replacing ViViT with Hiera demonstrates improvements on the fine-grained spatiotemporal reasoning benchmark, Perception Test, with accuracy gain of 1.29%. Similarly, adding Hiera yields an improvement on ActivityNet, achieving a 0.36% increase in accuracy. However, on other benchmarks, the original MERV remains the strongest model. Overall, we observe no significant performance improvement when training with Hiera, which aligns with expectations, since Hiera is under the same paradigm as ViViT, functioning as a temporal expert trained on short videos. We also hypothesize that Hiera is more sensitive to the temporal stride than ViViT, as ViViT can reasonably deduce motion from uniformly sampled frames. We expect performance to improve if we incorporate encoders trained on different paradigms and data sources or process a much greater number of frames simultaneously, which we will leave for future work. Nevertheless, we have demonstrated the easy extensibility of our proposed method to additional visual backbones.
>
> [2] Kim et. al., An Image Grid Can Be Worth a Video: Zero-shot Video Question Answering Using a VLM, 2024.
> [3] Xu et. al., SlowFast-LLaVA: A Strong Training-Free Baseline for Video Large Language Models, 2024.

---

> > ### Comment · Reviewer_6nZ3 · 2024-11-27
> >
> > I appreciate the authors' comprehensive responses and additional experiments. After reading the response and comments from other reviewers, my concerns about the novelty remain. While the paper provides empirical findings on the selection of image/video encoders and the fusion of multiple encoders to enhance video understanding tasks, it appears to be an incremental study. The underlying mechanism of this work exhibits minimal distinction from model ensembling and previous image-based MLLMs (Prismer, Vcoder, etc) with multiple encoders.

---

> > > ### Author Response · Authors · 2024-11-29
> > > **Thank you for your response!**
> > >
> > > Dear Reviewer 6nZ3,
> > >
> > > Thank you for your prompt response! We're glad to hear that all concerns have been satisfactorily resolved, except for the one regarding the novelty of our paper. However, we respectfully disagree that our work is incremental and the underlying mechanism exhibits minimal distinction from model ensembling and previous image-based MLLMs (Prismer, VCoder, etc.) with multiple encoders.
> > >
> > >
> > > We thank Reviewer 6nZ3 for bringing VCoder to our attention! VCoder is an MLLM for images that incorporates additional perception modalities, such as segmentation or depth maps, alongside RGB input. **Unlike Prismer and VCoder, which focus exclusively on image-based tasks and the integration of multiple perception modalities, our work represents the first VideoLLM to date that targets the effective utilization of multiple RGB-based visual and video encoders to achieve efficient and enhanced video understanding and reasoning.**
> > >
> > >
> > > We would like to recap our responses to your original disagreement of the novelty of our work compared to previous model ensembling and image-based works.
> > >
> > >
> > > It is important to note that **integrating multiple perception modalities into MLLMs is orthogonal to our research focus**. Prismer uses multiple modalities, while MERV is specifically on RGB inputs. Our primary aim is to explore optimal methods for leveraging multiple encoders. Our work provides a comprehensive study on utilizing multiple image (and video) encoders and examines how to efficiently and effectively integrate them, such as the spatio-temporal alignment of these features which is not present in previous works, thereby opening avenues for future research to build upon our findings. This foundation could significantly benefit research directions pursued by models like Prismer and VCoder. For instance, VCoder relies on a single visual encoder (CLIP) to process its three target perception modalities. Future versions of VCoder could integrate our image encoder choices and multi-encoder integration findings to process these different perception modalities to potentially achieve better performance.
> > >
> > >
> > > Furthermore, our work delivers newer, deeper and more distinct insights into multi-encoder feature integration strategies. In contrast, VCoder employs a relatively simple approach: a two-layer MLP maps CLIP-encoded segmentation map features into the LLM space, after which the multimodal tokens are concatenated and fed into the LLM. However, our findings demonstrate that this straightforward method results in lower average accuracy and higher computational costs compared to our carefully chosen design. Additionally, as mentioned in our previous response, our findings suggest that the Perceiver Resampler used by Prismer is not a limited choice for multi-encoder representation learning in videos. Therefore, **our work offers unique, novel, and critical insights into multi-encoder integration for VideoLLMs, supported by extensive analysis, baselines, and fair experiments. Thanks to deeper exploration, our findings diverge from the conclusions reached by Prismer and VCoder**, particularly regarding the integration strategy.
> > >
> > >
> > > We would love to address your concerns in a satisfactory manner, but we don’t know which specific points are not addressed to your liking, so your clarifications to this point would be of utmost importance. In our opinion, we believe these above distinctions underscore the significance of our contributions to advancing the field. We would be pleased to engage in further discussions should you have any remaining concerns.
> > >
> > >
> > > Sincerely,
> > > Submission #545 Authors

---

### Official Review · Reviewer_gpAa · 2024-11-09

**Soundness:** 3
**Presentation:** 4
**Contribution:** 3
**Rating:** 8
**Confidence:** 5

**Summary:**

This paper focuses on the vision encoder designs of VideoLLMs and proposes MERV (Multi-Encoder Representation of Videos) that utilizes multiple visual encoders to enhance the VideoLLMs' capabilities. MERV utilizes a spatial expert (DINOv2), a temporal expert (ViViT), an image-language contrastive expert (SigLIP), and a video-language contrastive expert (LanguageBind) as mixed video encoders and designs a pre-fusion projection to align their embedding size, then use cross-attention to perform spatial-temporal fusion as LLM's input. Extensive experiments demonstrate MERV's effectiveness and efficiency on multiple benchmarks.

**Strengths:**

1. This paper is well-written and easy to follow. The paper is well-motivated and the discussion of related works is clear, the method and experiment sections provide details for the proposed methods.
2. The proposed method MERV effectively combines multiple visual encoders to capture a wider range of video understanding capabilities is a significant advancement. The experiments and comparisons on multiple benchmarks show the effectiveness of such multi-encoder structures. Meanwhile, MERV introduces minimal extra parameters and computational overhead compared to existing single-encoder approaches, making it more suitable for practical use.
3. The paper includes well-conducted ablation studies on feature fusion strategies, pre-fusion projectors, and encoder combinations, providing insight into the effectiveness of design choices.

**Weaknesses:**

1. This paper shows that combining 4 typical encoders improves the capability of the VideoLLMs, I'm wondering if such gain from integrating more encoders further exists when more encoders are utilized.
2. It would provide more interpretability if the authors could do some analysis on how is each encoder chosen and utilized by the fusion module in some typical tasks.

**Questions:**

Please see the weakness section.

---

> ### Author Response · Authors · 2024-11-22
> **Response**
>
> Dear Reviewer, Thank you for giving us your time and energy for the thorough and insightful review of our paper. We are encouraged that you appreciate the importance of our motivation, the in-depth ablation studies across our settings, and the clear presentation of method and results. Please see our responses to the weaknesses you pointed out below:
>
> **W1: Does MERV still improve when we use more encoders?**
>
> Thank you for the interesting question. In response to both your question and Reviewer RqfQ’s question about using more modern video encoders, we conducted experiments using a newer SoTA video model Hiera-B+ [1], i.e. MViTv3, in both a four-encoder setting replacing ViViT and a five-encoder setting. The checkpoint we use was pretrained using the Masked Autoencoder (MAE) self-supervised learning technique on Kinetics 400 (K400), and then finetuned on K400 using supervised learning. On the other hand, ViViT was initialized from an ImageNet-21K/JFT backbone, and trained on K400 using supervised learning, so their training setups are similar.
>
> The results are presented in the table below, with the best result in **bold** and the second-best in *italics*. Replacing ViViT with Hiera demonstrates improvements on the fine-grained spatiotemporal reasoning benchmark, Perception Test, with accuracy gain of 1.29%. Similarly, adding Hiera yields an improvement on ActivityNet, achieving a 0.36% increase in accuracy. However, on other benchmarks, the original MERV remains the strongest model. Overall, we observe no significant performance improvement when training with Hiera, which aligns with expectations, since Hiera is under the same paradigm as ViViT, functioning as a temporal expert trained on short videos. We also hypothesize that Hiera is more sensitive to the temporal stride than ViViT, as ViViT can reasonably deduce motion from uniformly sampled frames. We expect performance to improve if we incorporate encoders trained on different paradigms and data sources or process a much greater number of frames simultaneously, which we will leave for future work.
>
> | Method | MSVD -Acc | MSVD -Score | MSRVTT -Acc | MSRVTT -Score | TGIF -Acc | TGIF -Score | ActivityNet -Acc | ActivityNet -Score | Perception Test -Acc |
>  | --- | --- | --- | --- | --- | --- | --- | --- | --- | --- |
>   | VideoLLaVA | 67.74 | 3.69 | 56.90 | 3.18 | 47.99 | 3.17 | 47.08 | 3.27 | 44.22 |
>  | MERV | **70.97** | **3.76** | **59.03** | **3.25** | **51.1** | **3.26** |  *50.87* | **3.34** | 46.21 |
> | MERV,  ViViT replaced with  Hiera | *69.68* | *3.74* | 57.64 | 3.22 | *50.38* | *3.24* | 50.24 | **3.34** | **47.50** |
>  | MERV + Hiera | 69.67 | 3.72 | *58.26* | *3.23* | 50.32 | 3.22 | **51.23** | **3.34** | *46.23* |
>
> [1] Ryali et. al., Hiera: A Hierarchical Vision Transformer without the Bells-and-Whistles, ICML 2023.
>
> **W2: Is there a pattern to how the feature fusion chooses encoders for certain tasks?**
>
> This is something we wondered as well, as we were hoping that feature fusion could do a selection of encoders. Our analysis found that the model instead is finding the best weights to perform a general unification of the features, rather than learning large changes in weights for different classes of examples. This is understandable given the objective, but also that individual videos can give rise to many different tasks requiring different skills, and a priori of the task, the model simply tries to learn the best representation for all potential questions.
>
> Nevertheless, we can still look at specific videos for which the feature fusion module tends to rely more heavily on particular encoders, and look for trends among those. In this setting, we checked the attention weights of all the videos in 4 of our benchmark datasets (MSRVTT, MSVD, TGIF, and Perception Test). For each encoder, we visualized the videos which give the highest attention weight for that specific encoder. The qualitative results are added in the appendix of the paper (Figure 12). As expected, ViViT attention weights are highest on videos with large motion, as ViViT has strong temporal motion understanding. Meanwhile, SigLIP, which is vision-language contrastively trained, attends more to videos that have textual data in the video. DINOv2 and LanguageBind are both preferred by videos with static scenes, but LanguageBind, being trained on videos, is preferred by videos with some foreground motion.

---

> > ### Comment · Reviewer_gpAa · 2024-11-26
> >
> > Thank you for the response. I've read them and i think my questions are well answered. I'll keep my rate as accept

---

> > > ### Author Response · Authors · 2024-11-29
> > > **Thank you for your response!**
> > >
> > > Dear Reviewer gpAa,
> > >
> > > We are thrilled to hear that all questions are well answered! Once again, thank you for your time and effort in reviewing our paper! We are immensely grateful for your support in advancing our work and helping us share it with a broader audience!
> > >
> > > Sincerely,
> > > Submission #545 Authors

---

### Author Response · Authors · 2024-11-22
**General Response**

Dear Reviewers, we thank you for spending your time and effort on reviewing our work. The comments and insights that you have provided are valuable, and we thank you for all the strengths and weaknesses that you have pointed out. We have addressed all of your concerns through individual responses, but we want to point out small modifications that we made from addressing your reviews.

* Originally we had called our model MERV and a variant of it that included finetuning through the LLM in stage 1 to be MERV-full. But this caused some confusion so we’ve renamed the two instantiations to be MERV (frozen) and MERV (full) in the new draft to explicitly disambiguate the two training recipes. In the responses below though, we continue to use the original naming convention to align with the reviewers’ original questions, so we use MERV to refer to MERV (frozen) (lines 240-248 in the revised draft)
* We have added Figure 3 to show the runtime of MERV, showing that including additional encoders only incurs minimal steptime overhead (line 400 in the revised draft).

We thank again for the support and insights that you have provided us.

---

### Meta-Review · Area_Chair_WWy3 · 2024-12-23

**Metareview:**

The submission addresses the problem of visual encoder design in video-language models. The proposed solution is to fuse multiple image and video encoders using cross attention. Extensive experiments were performed to demonstrate the effectiveness of the proposed approach on MSRVTT, TGIF, Something-Something-V2, among several other benchmarks. The submission received mixed ratings after rebuttal, including one accept (8), one borderline accept (6), and two borderline rejects (5). Below I summarize the main strengths and limitations of the submission, according to the reviews (after rebuttal discussion) and my own reading of the submission:

*Strengths:*
- The submission is very well motivated and clearly written. Reproducibility is high.
- The empirical evaluations are nicely designed, thorough, and the performances are strong.
- The reviewers especially appreciate that the submission presents comparisons with comparable training data and recipes.

*Weaknesses:*
- The proposed visual encoder fusion module (based on cross attention) has limited novelty, and the improvements over even simpler baselines (e.g. concatenation) appears to be non-significant.

The AC acknowledges that the submission offers a timely and valuable empirical study for the visual encoder design of video-language models. However, the AC also shares similar concerns with reviewers 6nZ3, ccLB, and RqfQ on the "technical novelty" aspect of the submission. While such "technical novelty" is not always required for a high-quality ICLR paper, the AC has reservations on: (1) how the proposed approach offers additional insights on top of relevant work such as Prismer, Cambrian and VCoder, the authors' argument that the proposed approach being the first to use multiple *RGB-based* visual and video encoders appears to overly emphasize on implementation details; (2) how the proposed approach and the empirical observations could inform and inspire visual-language understanding as a field in general. As a result, while the AC highly appreciates the strengths of the submission, they cannot recommend the submission to be accepted by ICLR 2025.

**Additional Comments On Reviewer Discussion:**

The authors did a solid job addressing the following questions:
- The choice of visual encoders, and if adding more encoders further improve the performance
- Additional analysis of the empirical results
- The choice of ViViT as the video encoder
- Inference time when using multiple encoders

After the discussion phase, reviewers ccLB and 6nZ3 have remaining concerns on the novelty of the proposed approach and the significance of performance improvement of cross attention over concatenation. Although the authors provided follow up responses on these questions (which the two reviewers did not respond to), the AC believes they are not sufficient to address the remaining concerns.

---

### Decision · Program_Chairs · 2025-01-22

Reject